# Severe Neuro-COVID is associated with peripheral immune signatures, autoimmunity and neurodegeneration: a prospective cross-sectional study

Manina M. Etter [1], Tomás A. Martins [1], Laila Kulsvehagen [2], Elisabeth Pössnecker [2], Wandrille Duchemin[3], Sabrina Hogan[1], Gretel Sanabria-Diaz[4], Jannis Müller [4,5], Alessio Chiappini[6], Jonathan Rychen[6], Noëmi Eberhard[6], Raphael Guzman[6,7], Luigi Mariani[6,7], Lester Melie-Garcia[4,5], Emanuela Keller[8], Ilijas Jelcic [9], Hans Pargger [10], Martin Siegemund [10], Jens Kuhle[5], Johanna Oechtering[4,5], Caroline Eich[1], Alexandar Tzankov [11], Matthias S. Matter [11], Sarp Uzun[11], Özgür Yaldizli[4], Johanna M. Lieb [12], Marios-Nikos Psychogios[12], Karoline Leuzinger [13,14], Hans H. Hirsch[14,15,16], Cristina Granziera[4,5], Anne-Katrin Pröbstel [2] & Gregor Hutter [1,6,7] ✉

Growing evidence links COVID-19 with acute and long-term neurological dysfunction. However, the pathophysiological mechanisms resulting in central nervous system involvement remain unclear, posing both diagnostic and therapeutic challenges. Here we show outcomes of a cross-sectional clinical study (NCT04472013) including clinical and imaging data and corresponding multidimensional characterization of immune mediators in the cerebrospinal fluid (CSF) and plasma of patients belonging to different Neuro-COVID severity classes. The most prominent signs of severe Neuro-COVID are blood-brain barrier (BBB) impairment, elevated microglia activation markers and a polyclonal B cell response targeting self-antigens and non-self-antigens. COVID-19 patients show decreased regional brain volumes associating with specific CSF parameters, however, COVID-19 patients characterized by plasma cytokine storm are presenting with a non-inflammatory CSF profile. Post-acute COVID-19 syndrome strongly associates with a distinctive set of CSF and plasma mediators. Collectively, we identify several potentially actionable targets to prevent or intervene with the neurological consequences of SARS-CoV-2 infection.

The prevalence of neurological symptoms (NS) after severe acute respiratory syndrome coronavirus 2 (SARS-CoV-2) infection, altogether termed "Neuro-COVID", differs significantly between studies and can rarely be explained by direct virus effects[1,2]. In support of a detrimental immune response, neuropathological evidence of hyperactive microglia has been provided[3], and postmortem studies postulate activated microglia as a dominant immune cell population in coronavirus disease 2019 (COVID-19) brains. In autoptic single cell RNA sequencing studies, COVID-19 brains display dysregulated astrocytic and microglial signatures[4], accompanied by deranged

choroid plexus cell types[5,6]. Additionally, the formation of microglia and T-cell nodules were detected across brain compartments as a site of greatest T cell and microglia activation[1]. Accordingly, specific immune alterations in the cerebrospinal fluid (CSF) of Neuro-COVID patients featured an increase of exhausted T cells, probably due to repetitive stimulation. Hence, these observations indicate a compromised antiviral response pointing towards immune-mediated mechanisms responsible for severe Neuro-COVID[2,7].

There is strong evidence of brain-related pathologies in COVID-19. Schwabenland et al.[1] confirmed the presence of amyloid precursor protein deposits in COVID-19 brains, suggesting axonal damage as a result of immune activation. Accordingly, Douaud et al.[8] identified a reduction in gray matter thickness in primary olfactory cortex regions, and a reduction in brain size in a longitudinal study of COVID-19-related brain pathologies. However, SARS-CoV-2 RNA has rarely been detected in the CSF of COVID-19 patients, even in those displaying NS[2,9,10]. Moreover, new-onset humoral autoimmunity, including antineuronal antibodies, in COVID-19 individuals has been observed, even in the absence of increased conventional inflammatory CSF parameters and lacking evidence of inflammation upon neuroimaging[11,12]. Yet, it still remains controversial whether these alterations represent specific central nervous system (CNS) infection or are bystander effects of systemic COVID-19.

Here, we perform an in-depth characterization of immune mediators in the CSF and plasma of clinically well-characterized Neuro-COVID patients and correlate these findings with brain imaging data and a 13-month follow-up. A vigorous microglia reactivity, a dysfunctional blood-brain barrier (BBB) and CNS ingressing B cells mainly characterized severe Neuro-COVID. We observe a plasma cytokine storm combined with a non-inflammatory CSF profile, even in severe Neuro-COVID. However, particular CSF and plasma inflammatory parameters are associated with decreased regional brain volumes in COVID-19 patients and post-acute COVID-19 syndrome. These findings may be addressed to prevent COVID-19 related neurological impairment in the future.

## Results
### Clinical characteristics of the study cohorts and study interventions
In total, we screened 310 patients in order to reach a target study population of 40 (mean [SD] age, 54 [20] years; 17 women [42%]) participants (Fig. 1a). One patient did not meet our inclusion criteria (≥18 years, non-pregnant, positive PCR test) due to pregnancy. The other 269 screened patients declined participation, mostly because of fear of lumbar puncture (LP) side effects. In 5/40 (12.5%) cases, LP failed to deliver sufficient CSF amounts, precluding further downstream analysis. Within these patients, 1 suffered from multiple sclerosis (MS), which in consequence did not influence our CSF analysis.

Patient characteristics, main NS and follow-up details per severity class are summarized in Table 1. COVID-19 patients (n = 40; mean [SD] age, 54 [20] years; 17 women (42%)) were clinically classified into absent or mild (n = 18), moderate (n = 7) and severe (n = 15) Neuro-COVID classes I, II and III based on the severity of their NS at presentation (Fig. 1a, Table 1). Neurological symptom severity was assessed (1) according to Fotuhi et al.[13] and (2) on our clinical experience. Class I was defined by absent/mild NS, including headache, dizziness, anosmia and ageusia. Class II encompassed moderate NS, including fatigue, acute peripheral neuropathy and myopathy, whereas class III was specified by severe NS, including seizures, stroke or intracranial hemorrhage, encephalopathy, coma or death. Additionally, we assessed COVID-19 severity (not focusing on NS) using the WHO clinical progression score[14]. Accordingly, patients' clinical COVID-19 severity was scored from 0 (uninfected) to 10 (dead, Table 1).

Control groups consisted of biobanked, age- and sex-matched paired CSF and plasma samples from non-MS inflammatory control disorders ("CNS inflammatory controls") (n = 25; mean [SD] age, 54 [19] years; 12 women [48%]) and healthy donors (n = 25; mean [SD] age, 52 [18] years; 12 women [48%]). CNS inflammatory controls consisted mostly of infectious conditions, including herpetic infections, viral meningitis and meningoencephalitis, eosinophilic or tuberculous meningoencephalitis, neuroborreliosis, neurosarcoidosis, Susac's syndrome, and one patient with autoimmune encephalitis and Rasmussen's encephalitis (Supplementary Table 1).

Study interventions are summarized in Fig. 1b–d. Thirty-two patients underwent contrast-enhanced brain MRI imaging followed by standard and algorithm-based image analysis. However, not all patients (5 out of 40) were MRI scanned because of the complex logistics, staffing and medical surveillance issues during the COVID-19 pandemic. Therefore, 5 patients underwent cranial computed tomography (CT) instead, whereas one patient was imaged with both brain MRI and cranial CT. For volumetric imaging analysis, we created a modified cohort of COVID-19 patients (n = 35), consisting of 22 patients out of the main study cohort and additional 13 patients undergoing brain MRI during their COVID-19 infection.

### Class III patients have an impaired BBB and a polyclonal B-cell response
In class III patients, CSF protein and lactate levels were significantly increased compared to class I and II (Fig. 2a). In contrast, CSF leukocytes were not elevated in COVID-19 patients, which signified CNS inflammatory controls. Importantly, CSF glucose was increased even in class I and II versus (vs) healthy controls, and was a significant discriminator between class III and CNS inflammatory controls (Fig. 2a). However, the CSF/blood glucose ratio of class III was significantly higher in class III compared to class I/II patients, although diabetic patients were more prevalent in this group (class I: 11.1%; class II: 14.3%; class III: 53.3%, (Fig. 2a, Table 1).

Routine CSF parameters are described in Supplementary Table 2 and Supplementary Data 1. In tendency, the CSF/plasma albumin ratio (Fig. 2a), as well as the total CSF IgG levels (Supplementary Fig. 1b), were higher in class III patients.

In line with recent research[2], SARS-CoV-2 RNA was not detectable in the CSF. However, we were able to detect SARS-CoV-2 Spike (S) protein antibodies in 12 plasma and 3 CSF samples (Fig. 2c), yet the antibody index (AI) pointed to a peripheral synthesis of these intrathecal antibodies (Supplementary Data 1).

Next, we determined the presence of autoreactive antibodies in both CSF and plasma and represented them by non-metric multidimensional scaling (NMDS) plots. While the plasma antibody profiles of class III and CNS inflammatory controls segregated, their CSF profiles overlapped (Fig. 2b).

We could not detect reactivities against known CNS myelin antigens in the plasma (Supplementary Fig. 1a), but found elevated anti-dsDNA-IgG/IgA and anti-gut microbiota IgA responses in the CSF of class III compared to class I patients and also to CNS inflammatory controls (Fig. 2c). This was paralleled by an elevated anti-BSA reactivity. The AI pointed towards a peripheral antibody production. Plasma anti-dsDNA-, anti-BSA- and anti-gut microbiota responses did not differ significantly across groups (Supplementary Fig. 1b).

Of note, we identified similar total levels of plasma antibodies in all classes (Supplementary Fig. 1c), whereas class III patients depicted a significantly higher total antibody concentration in the CSF (Supplementary Fig. 1d).

To evaluate the B-cell response of patients having mild or severe symptoms, we sequenced Ig heavy chain CDR3 (complementarity determining region 3) in RNA isolated from peripheral blood mononuclear cells (PBMC) of four class I and class III patients each (Supplementary Table 3). First, we found a significantly higher number of plasma B-cell clones in class I compared to class III. The number of B-cell clones correlated with increased plasma IgG levels

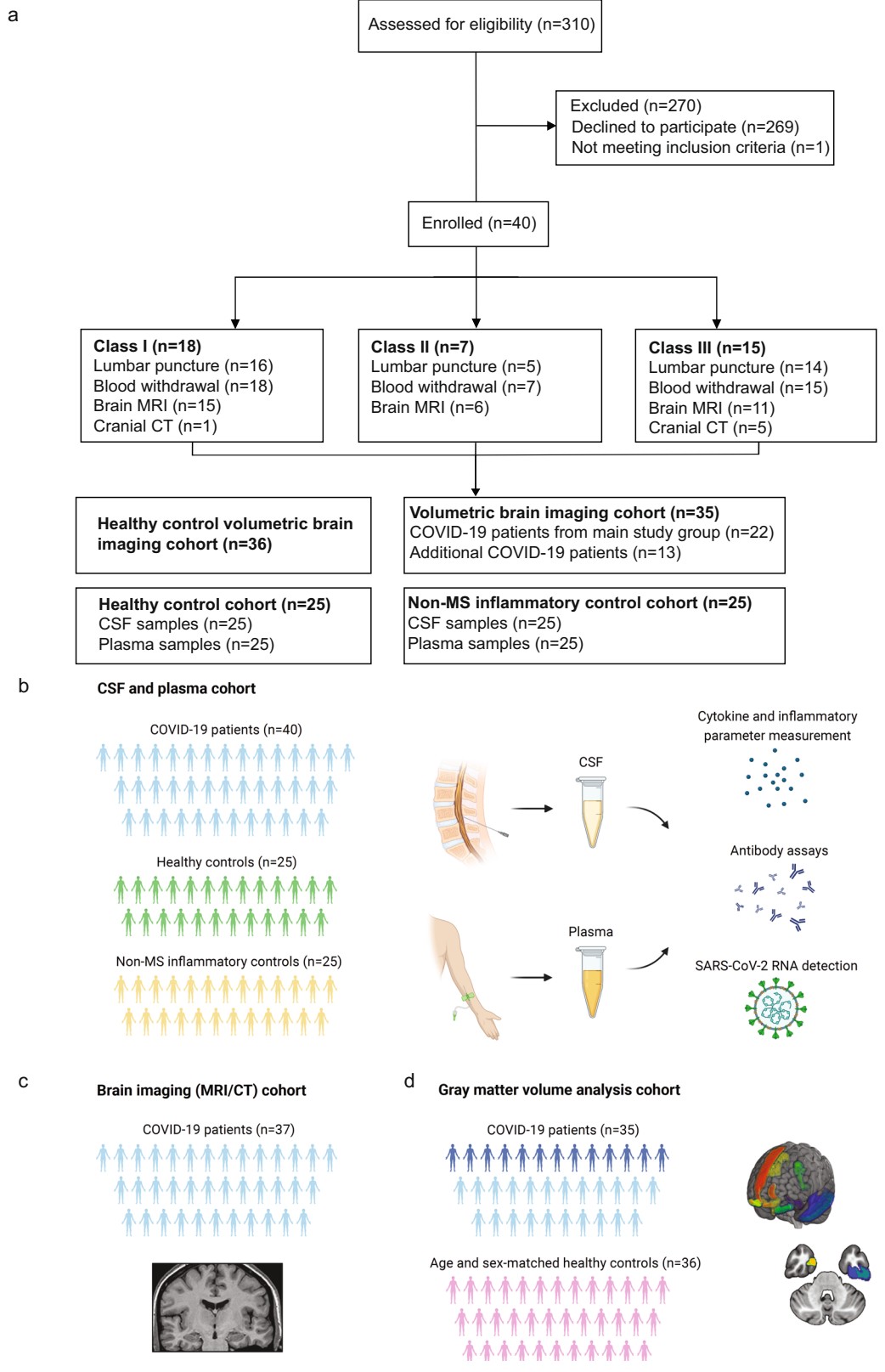

suggesting that the number of unique B cells in the blood might be an indicator of high IgG production (Fig. 3a). There was no significant difference in Shannon diversity and evenness between cohorts[15]. Both groups had an evenness value close to 1 indicating a polyclonal distribution of B-cell clones[15] (Fig. 3b). Indeed, only two patients from class I had B-cell clones accounting for more than 5% of all B-cell clones. In contrast, all class III patients showed a more polyclonal distribution without an expanded high frequency B-cell clone (Fig. 3c). B-cell clones of both class I and class III patients showed a similar Gaussian CDR3 length distribution with a similar mean CDR3 length (Fig. 3d). Taken together, BCR sequencing analysis revealed a more specific antibody response in class I patients, whereas in class III patients the response was more diverse due to polyclonality.

**Fig. 1 | CONSORT diagram and schemes illustrating the project design.**
**a** Consort flow diagram. Patients who tested positive for SARS-CoV-2 were assessed for eligibility (n = 310), of which 269 declined to participate and 1 failed to meet inclusion criteria. Enrolled patients (n = 40) were allocated to different severity classes of Neuro-COVID according to Fotuhi et. al.[13] with 18 in class I, 7 in class II and 15 in class III. *Schemes illustrating the study design:* **b** Paired cerebrospinal fluid (CSF) and plasma samples were collected from 40 COVID-19 patients. Paired CSF and plasma samples from healthy (n = 25) and non-MS inflammatory neurological disease controls (n = 25) were retrospectively obtained. **c** In 37 of the COVID-19

patients, a contrast-enhanced MRI or CT scan was conducted and evaluated by a board-certified neuroradiologist. **d** Brain volumetric analysis was performed in 35 COVID-19 patients. This cohort included 22 patients of the main study cohort from whom Magnetization prepared–rapid gradient echo (MPRAGE) pulse sequences and paired CSF and plasma samples were obtained (*light blue*) and an additional 13 patients who underwent brain MRI during COVID-19 infection (*dark blue*). A cohort of 36 healthy age and sex-matched individuals served as the control group. **b**–**d** Created with Biorender.com.

## Table 1 | Characteristics of COVID-19 patients

| | Class I (n = 18) | Class II (n = 7) | Class III (n = 15) |
|---|---|---|---|
| **Age, years, mean (SD)** | 48 (21) | 49 (19) | 62 (17) |
| Range, years | 22–80 | 23–73 | 22–98 |
| **Sex** | | | |
| Female, n (%) | 8 (44.4%) | 3 (42.9%) | 6 (40.0%) |
| Male, n (%) | 10 (55.6%) | 4 (57.1%) | 9 (60.0%) |
| **Delay, d, mean (SD)** | | | |
| Diagnosis to LP | 3 (3) | 3 (2) | 4 (4) |
| Range, d | (0–8) | (1–5) | (0–11) |
| Diagnosis to blood withdrawal | 3 (3) | 3 (2) | 4 (4) |
| Range, d | (0–8) | (0–7) | (0–15) |
| Diagnosis to MRI/CT | 4 (3) | 4 (4) | 6 (7) |
| Range, d | (1–11) | (0–11) | (0–12) |
| **Past medical history** | | | |
| Arterial hypertension, n (%) | 4 (22.2%) | 1 (14.3%) | 7 (46.7%) |
| Type 2 diabetes, n (%) | 2 (11.1%) | 1 (14.3%) | 8 (53.3%) |
| Dyslipidemia, n (%) | 4 (22.2%) | 0 (0%) | 2 (13.3%) |
| Chronic kidney disease, n (%) | 2 (11.1%) | 0 (0%) | 4 (26.7%) |
| Coronary artery disease, n (%) | 4 (22.2%) | 1 (14.3%) | 1 (6.7%) |
| Cancer of any type, n (%) | 0 (0%) | 1 (14.3%) | 2 (13.3%) |
| COPD, n (%) | 0 (0%) | 0 (0%) | 0 (0%) |
| **Pre-existing neurological disorder** | | | |
| Multiple sclerosis, n (%) | 1 (5.6%) | 0 (0%) | 0 (0%) |
| Underwent LP, n | 0 | – | – |
| Underwent blood withdrawal, n | 1 | – | – |
| Myasthenia gravis, n (%) | 0 (0%) | 1 (14.3%) | 0(0%) |
| Underwent LP, n | – | 0 | – |
| Underwent blood withdrawal | – | 1 | – |
| **Main neurological symptom/syndrome** | | | |
| Headache/Dizziness, n (%) | 11 (61.1%) | 0 (0%) | 0 (0%) |
| Loss of smell/taste, n (%) | 7 (38.9%) | 0 (0%) | 0 (0%) |
| Acute peripheral neuro-pathy, n (%) | 0 (0%) | 2 (28.6%) | 0 (0%) |
| Myopathy, n (%) | 0 (0%) | 5 (71.4%) | 0 (0%) |
| Seizures, n (%) | 0 (0%) | 0 (0%) | 1 (6.7%) |
| Cerebrovascular disease, n (%) | 0 (0%) | 0 (0%) | 2 (13.3%) |
| Encephalopathy, n (%) | 0 (0%) | 0 (0%) | 12 (80%) |
| **Clinical evolution** | | | |
| ICU, n (%) | 0 (0%) | 0 (0%) | 11 (73.3%) |
| Hospital ward, n (%) | 13 (72.2%) | 7 (100%) | 4 (26.7%) |
| Outpatient clinics, n (%) | 5 (27.8%) | 0 (0%) | 0 (0%) |
| **WHO clinical progression scale (0–10), mean** | 3.2 | 3.9 | 8.2 |
| Range | 2–5 | 2–5 | 4–10 |

Demographics, outcomes, clinical and paraclinical characteristics of different Neuro-COVID class patients (n = 40). Patients could have more than 1 pre-existing illness in past medical history.
*SD* standard deviation, *LP* lumbar puncture, *MRI* magnetic resonance imaging, *CT* computed tomography, *COPD* chronic obstructive pulmonary disease, *CSF* cerebrospinal fluid.

## Targeted proteomic analysis of CSF and plasma reveals a robust peripheral immune response in Neuro-COVID and a class III-specific signature

To further elucidate mechanisms associated with Neuro-COVID, and to identify class-specific secretome patterns, we performed targeted soluble CSF and plasma proteomics. We identified predominant plasma secretion of a large number of soluble proteins in Neuro-COVID class III patients compared to controls (Fig. 4a, Supplementary Fig. 2a), suggesting a Neuro-COVID class-dependent plasma signature. Class I and II patients had an increased plasma secretome compared to controls, in line with the previously described peripheral cytokine storm in COVID-19[16]. The CSF protein pattern was different: while class I and II depicted relatively similar profiles as healthy controls, a Neuro-COVID class III-specific signature with differences to CNS inflammatory controls emerged (Fig. 4a, Supplementary Fig. 2b). Notably, CSF total protein levels progressively increased from class I to III, indicating a correlation between CSF proteomics and NS (Fig. 4a).

In non-metric multidimensional scaling (NMDS) plots of plasma proteins, class II and III segregated from healthy and CNS inflammatory controls, whereas the secretome in class I partially overlapped with class II and III and also the control cohorts (Fig. 4b, Supplementary Data 2). In the CSF, protein patterns did mostly overlap and a clear segregation was not observable. However, class III patients tended to display a similar pattern as CNS inflammatory controls, while, in tendency, healthy controls, class I and class II patients showed a likewise pattern diverging from class III and CNS inflammatory controls (Fig. 4b, Supplementary Data 2).

Next, we investigated the relative concentration of each molecule between CSF and plasma ($\log_2$ normalized CSF/plasma index) to ascertain whether they result from intrathecal or peripheral synthesis. We found that most proteins were intrathecally (CSF/plasma ratio >0) secreted in CNS inflammatory controls, whereas proteins in COVID-19 patients were mainly peripherally synthesized (CSF/plasma ratio <0). Of note, TRANCE/RANKL was the only intrathecally synthesized protein in class III compared to CNS inflammatory controls (Supplementary Fig. 3, *rose plots*, Supplementary Fig. 4, *heatmaps*, Supplementary Data 3, 4).

## Neuro-COVID class III features are manifestations of microglia regulation, neurodegeneration and BBB disruption

Next, we analyzed individual analytes across all study groups to decipher potential discriminative markers (Supplementary Data 5).

As reported previously, plasma IL-6, IL-8, EN-RAGE, HGF, VEGFA, PD-L1 and TNFRSF12A levels were associated with Neuro-COVID severity[17] and distinct from CNS inflammatory controls (Fig. 5a), which lacked peripheral inflammation. Furthermore, plasma TNFRSF11B, EZR and CCL23 were increased in class III vs. CNS inflammatory controls (Fig. 5a). In contrast, plasma levels of neurotrophic and neuroprotective factors such as BMP-4, CLEC10A, CNTN5, GDF-8, NTRK2, ROBO2 and GDNFRα3 were lower in class III compared to both CNS inflammatory controls and class I patients (Fig. 5b)[18–25]. Compared to CNS inflammatory controls, class III patients displayed higher plasma 4E-BP1 levels (Fig. 5c). HAGH was the only protein displaying higher plasma levels in class I vs. class III (Fig. 5d).

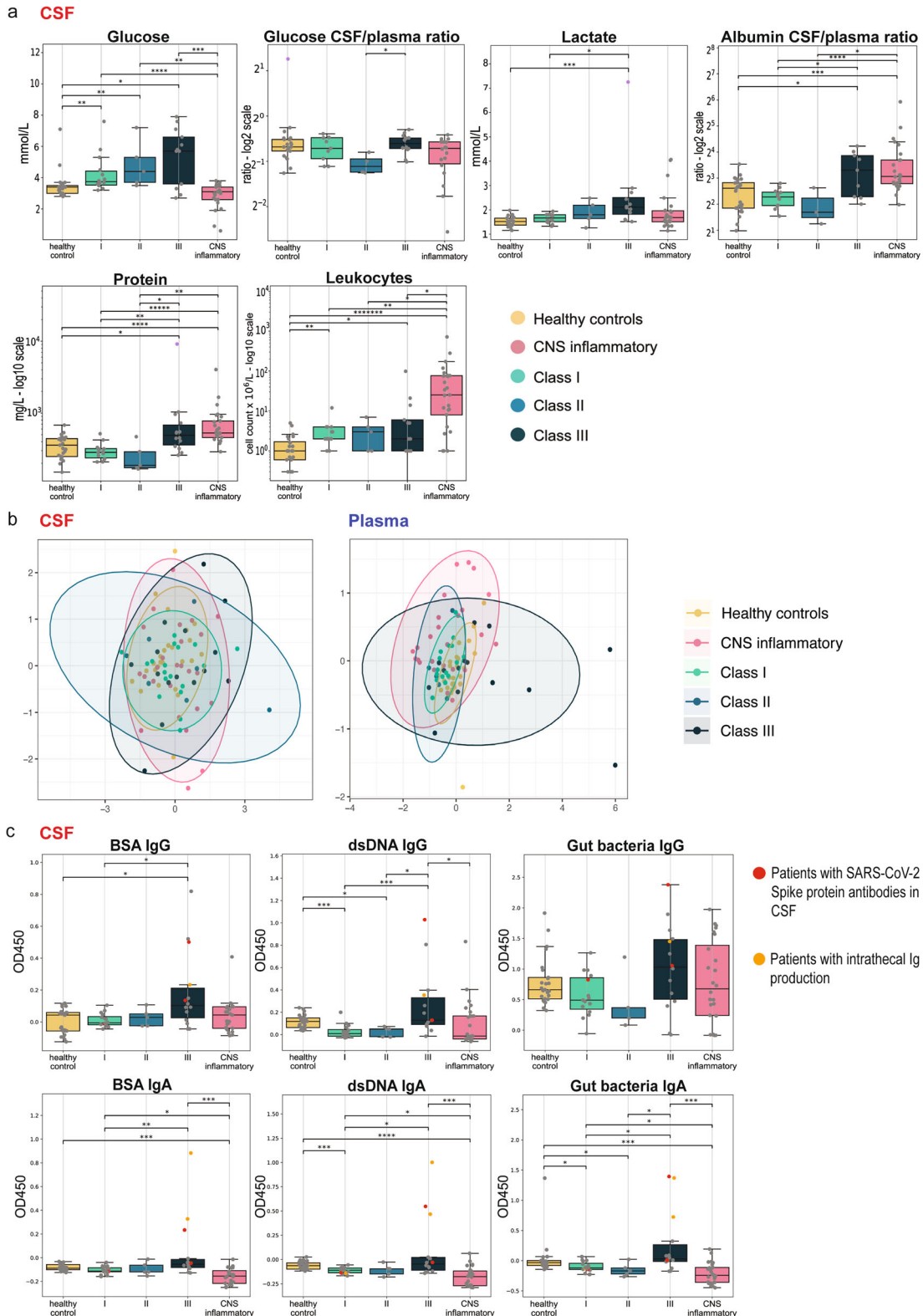

Several CSF soluble protein levels, particularly some deemed involved in microglia regulation, neurodegeneration and blood-brain barrier (BBB) disruption, including IL-8, MSR1, 4E-BP1, CD200R1, TNFRSF12A and EZR were increased in class III compared to class I (Fig. 5e)[18,19]. However, only TNFRSF11B levels were both discriminating class III from CNS inflammatory controls and gradually increasing among Neuro-COVID classes (Fig. 5f).

### Mediators involved in microglia regulation, tissue damage and blood−brain barrier disruption are associated with a high WHO clinical progression scale score

Next, we assessed the association of CSF/plasma proteins with COVID-19 severity[14]. We used a complement of four models, consisting of a backward-ordinal-, forward-ordinal-, best linear- and most-regularized ordinal models, and cross-referenced the results to provide a robust set of biomarkers associated with COVID-19 severity.

**Fig. 2 | Routine inflammatory CSF parameters and B-cell response in Neuro-COVID patients. a** Box plot representation of routine cerebrospinal fluid (CSF) parameters including glucose (mmol/L), glucose CSF/plasma ratio (log2 scale), lactate (mmol/L), albumin CSF/plasma ratio (log2 scale), total protein (mg/L, log10 scale) and leukocytes (cell count × $10^6$/L − log10 scale) (center line at the median, upper bound at 75th percentile, lower bound at 25th percentile) with whiskers at minimum and maximum values. Patients that were excluded from the analysis are indicated in violet. Statistics: The data for each parameter, except the leukocyte count, was marginalized on sex and age. Statistics: The data for each parameter was marginalized on sex and age. Two-sided Mann−Whitney-U test was applied, *p* value correction was performed with Benjamin-Hochberg (BH)-procedure (adjusted p: *<0.05; **<0.01; ***<0.001; if not otherwise indicated: not significant). **b** Non-metric multidimensional scaling (NMDS) plots of merged anti-BSA-, anti-dsDNA- and anti-gut bacteria antibodies in the CSF *(left plot)* and plasma *(right plot)* of each class. Data points are colored by category. Each point represents one patient. Patients with a more similar antibody composition are closer together and, conversely, those that were more dissimilar depict a greater distance. The ellipses represent the 95% confidence interval within subgroups. **c** Box plot representation of CSF levels (OD450; optical density at 450 nm) of anti-BSA, anti-dsDNA-and anti-gut bacteria (RePOOPulate)-IgG/IgA per patient and control group. Patients with anti-SARS-CoV-2 Spike protein antibodies in the CSF indicated in red, those with intrathecal IgG or IgA production in orange, respectively. Statistics: The data for each parameter except the IgG data was marginalized on sex and age. Two-sided Mann−Whitney-U test was applied, *p* value correction was performed with Benjamin-Hochberg (BH)-procedure (adjusted *p*: *<0.05; **<0.01; ***<0.001; if not otherwise indicated: not significant). Source data of (**a**, **c**) are provided as a Source Data file.

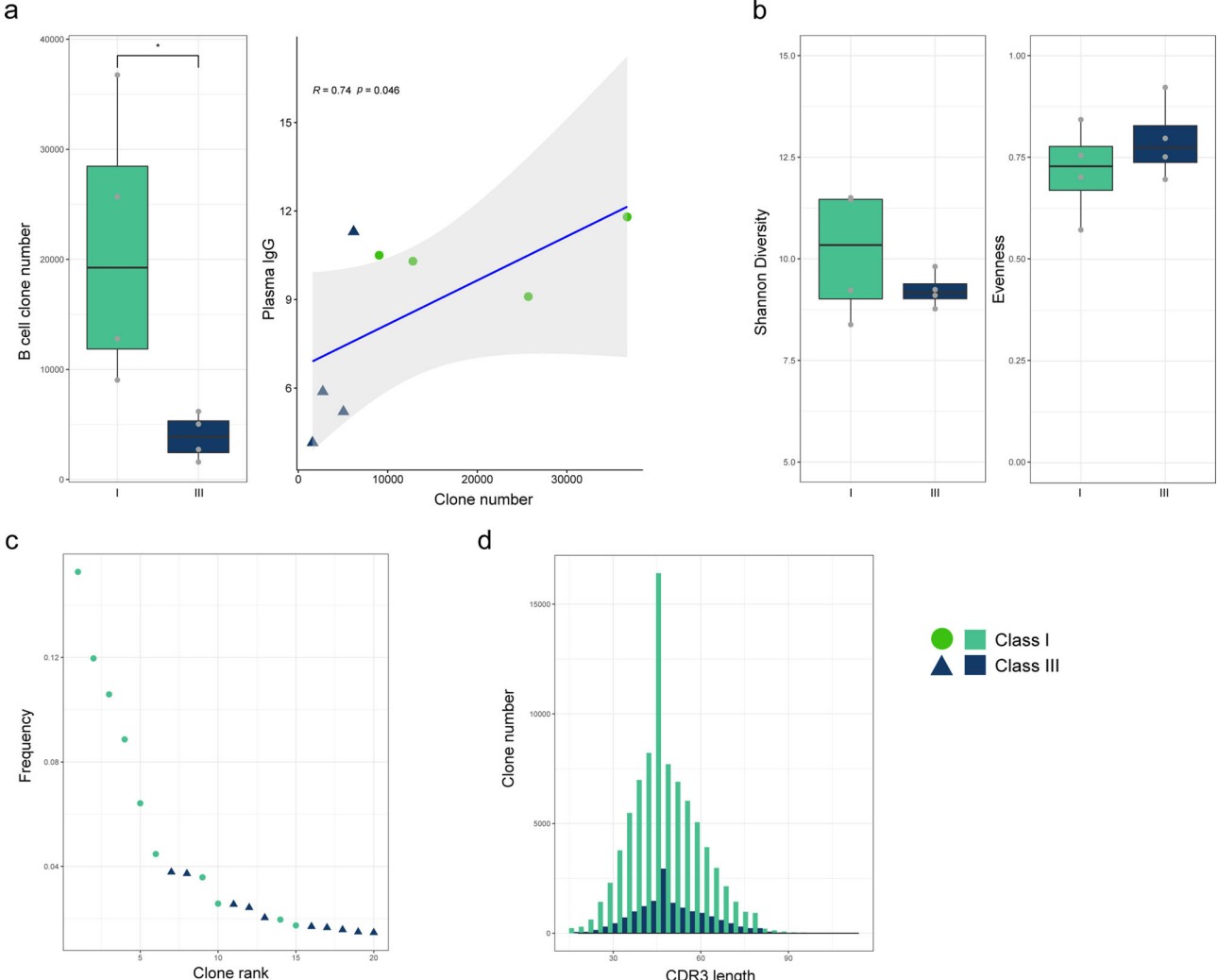

**Fig. 3 | B-cell receptor sequencing reveals a more specific antibody response and a higher B-cell clone number in class I patients, whereas class III patients depicted a more diverse response due to polyclonality. a** Box plot representation of B-cell clone number in class I (*n* = 4) and class III patients (*n* = 4) respectively. B-cell clone number was significantly higher in the blood of class I patients compared to class III (Mann−Whitney-U test: *p*: =0.02857, spearman's rank correlation test: *R* = 0.738, *p* = 0.04583). Class I patients (*n* = 4) had higher plasma IgG levels which correlated with the number of B-cell clones in the blood. **b** Box plot representation of Shannon Diversity and Evenness in class I (*n* = 4) and class III (*n* = 4) patients. Diversity analysis showed that there was no significant difference between class I and II. **c** Representation of all B-cell clones from each class clustered together and ranked according to their total frequency (relative abundance) in the immune repertoire of a patient. Then, top 10 highest frequency clones from each class selected, grouped together and ranked again according to the total frequency. As indicated in the dot plot, the clones making up more than 5% of BCR immune repertoire belonged to class I patients (*n* = 4). Only two B-cell clones of class III patients were in the top 10. **d** B-cell immune repertoire of both classes (class I and III patients: *n* = 8) showed a similar Gaussian distribution in CDR3 nucleotide length and same median CDR3 nucleotide length. Each bar represents the number of B cells having a specific CDR3 nucleotide length. Boxplots indicate median and lower (25th)−upper (75th) quartile and whiskers show the minimum-maximum values. Each dot represents individual samples. Statistics: Two-sided Wilcoxon rank-sum test and spearman's rank correlation were applied (adjusted *p*: *<0.05; if not otherwise indicated: not significant, *R* Spearman correlation coefficient). BCR B-cell receptor.

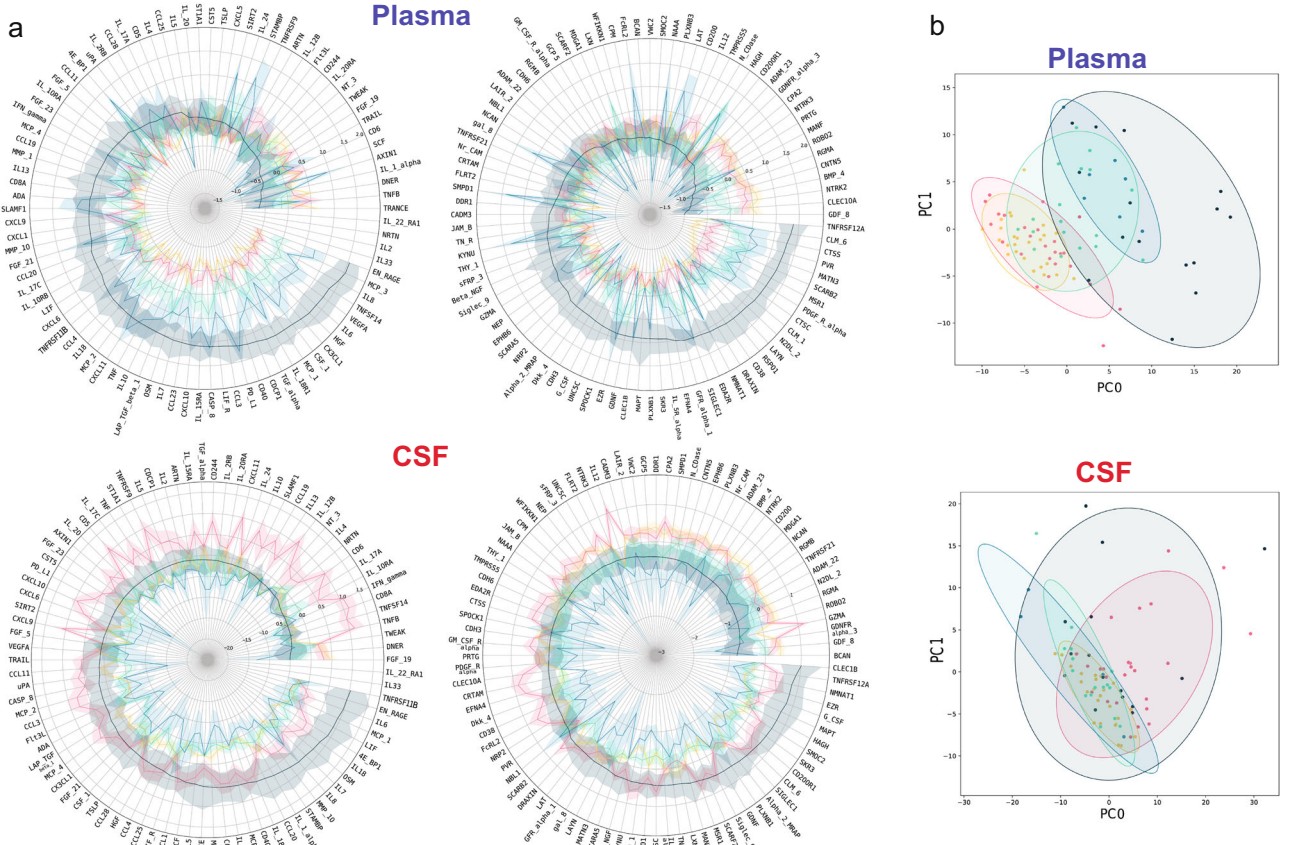

**Fig. 4 | Neuro-COVID patients display a vigorous peripheral immune response and specific CSF alterations including analytes with high predictive value for class III development and strong CSF-plasma correlation. a** Rose plots representing Z-scores of marginalized normalized protein expression (NPX) of 192 soluble proteins in CSF and plasma. For better visualization, analytes have been grouped into 'inflammatory' *(left panels)* and 'neurological' *(right panels)* proteins.

**b** Non-metric multidimensional scaling (NMDS) plots of 192 examined soluble proteins in CSF and plasma. Each patient is presented by one dot, and colored according to healthy controls, CNS inflammatory controls or Neuro-COVID class I-III. The ellipses represent the 95% confidence interval within subgroups. Source data are provided as a Source Data file.

Of note, several proteins with significantly higher CSF and plasma levels in class III Neuro-COVID patients were additionally associated with COVID-19 severity[14], emphasizing the association of these mediators with COVID-19 severity (Fig. 6a). High plasma levels of MSR1, as well as high CSF TNFRSF12A and IL-8 levels demonstrated the most robust association with COVID-19 severity[19,22,26,27]. Furthermore, plasma IL-8, IL-6, TNFRSF11B, and CSF EZR levels were allied to severe COVID-19 using the ordinal backward and best linear model[21,22,27–29], whereas plasma 4E-BP1 and CSF levels of PD-L1, BMP-4, CLEC10A and ROBO2 withstanded using one model only[30–34]. Taken together, mediators involved in pro-inflammatory cascades, BBB disruption, microglia and astrocyte activation, and tissue damage displayed the strongest association to COVID-19 severity.

## CSF-plasma correlations identify a neuronal damage signature in class III, encompassing predictive markers for severe Neuro-COVID

We then measured correlations between each CSF and plasma mediator, and ranked them by correlation strength. Since plasma samples are routinely obtained for diagnostics, we focused on predictive proteins with a strong CSF-plasma correlation to identify biomarkers associated with severe Neuro-COVID.

Assuming a cut-off of >0.45 in the Kendall-Tau correlation matrix, class-specific CSF-plasma correlations were noted and ranked (Supplementary Fig. 5a, *Venn diagram*). CNS inflammatory controls and healthy controls were characterized only by few strong CSF-plasma correlations compared to the Neuro-COVID groups.

We observed a gradual change in correlations from class I to class III. Only a few overlapping soluble proteins with strong correlations were detected, whereas 10-12 individual class-defining proteins were identified (Supplementary Fig. 5a, b, *Venn diagram and UpSet plot*). In class I, the strongest correlations (value >0.55) were characterized by a myeloid/eosinophil pro-inflammatory signature exemplified by SIGLEC1, MCP2, IL-8 and CLM1[20–23] (Supplementary Fig. 5c, *heatmap*). In class II, an activated T cell-mediated signature prevailed, defined by CCL25, CD8A, GZMA, TNFRSF9 and IL2-RB, while myeloid correlations overlapped between class I and II[24,25,35]. In class III, the pattern with the strongest CSF-plasma correlations shifted to a chronic inflammatory and neuronal damage signature encompassing CTSC, KYNU, TNFRSF12A, and CXCL9[18,19,36,37], potentially implicating T-cell exhaustion during disease progression, whereas an adequate T-cell function could be preserved in class II.

By ranking mediators based on their AUC-ROC score to discriminate class I and II vs. class III, nine analytes (4 CSF and 5 plasma proteins) displayed a score of >0.85, suggesting a high predictive power for class III development (Fig. 6b, Supplementary Data 6). Among these, TNFRSF12A additionally depicted a strong CSF-plasma correlation (class III: 0.56; class II: −0.4; class I: 0.2), validating it as a predictive biomarker for severe Neuro-COVID that could be routinely used in clinics.

Multivariate analysis revealed that plasma IL-8, TNFRSF12A, MCP-3, PVR, and CSF CD200R1 ranked high in both AUC-ROC and random forest importance scores (Fig. 6c).

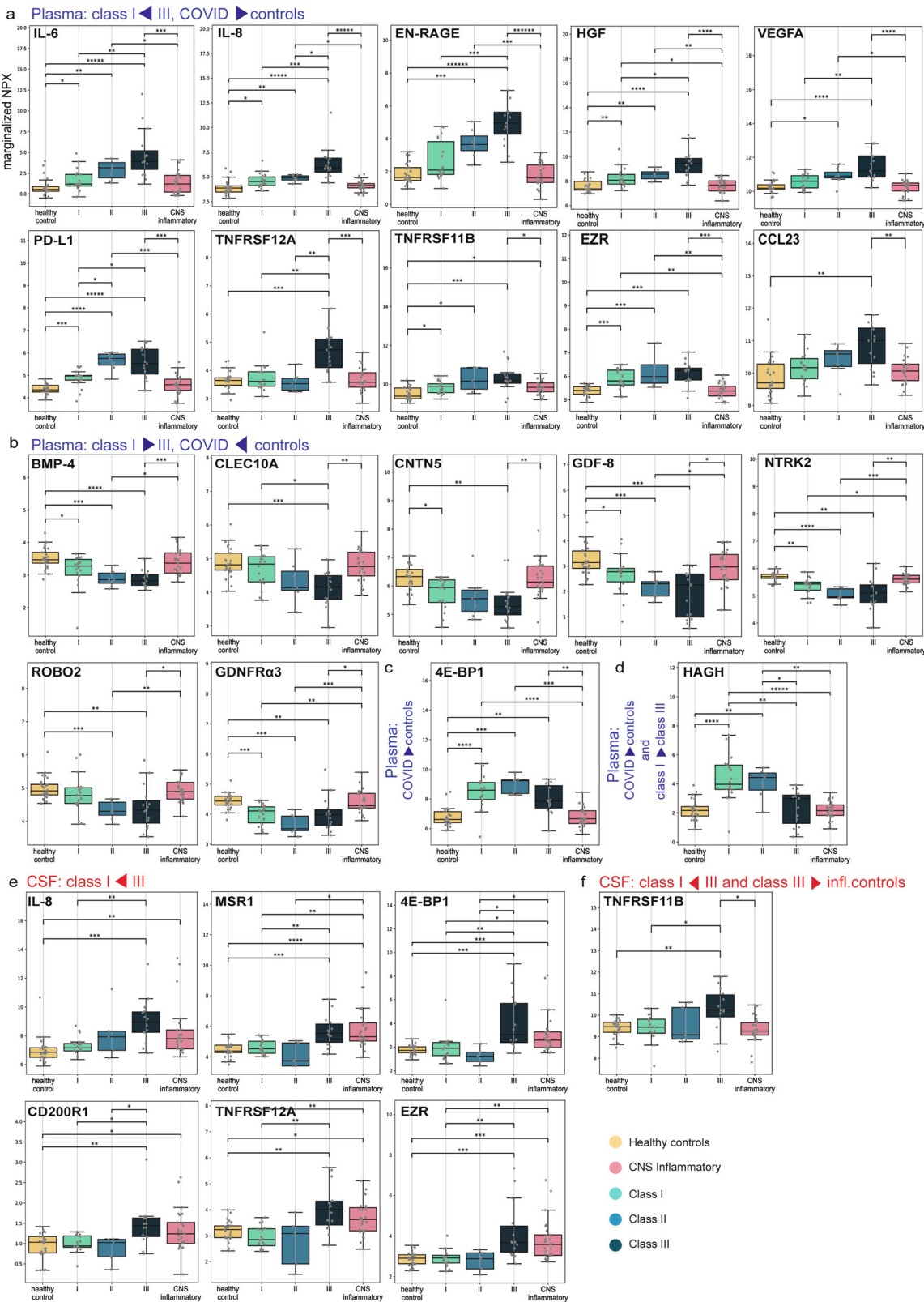

Plasma IL-8, CSF TNFRSF12A and EZR were strongly associated with both Neuro-COVID class III development and WHO COVID-19 severity, highlighting their importance as potential predictive biomarkers (Fig. 6d)[19,22,27–29]. The most likely immune cell sources of plasma and CSF proteins associated with both severe COVID-19 and severe Neuro-COVID development are illustrated in Supplementary Fig. 6.

**Neuro-COVID class III patients feature striking findings on brain imaging while most class I and II patients lack evidence of neuroinflammation**

Exemplary brain images of each class (obtained during COVID-19) are depicted in Fig. 7a–c. Detailed imaging findings are presented in Supplementary Data 7. The most frequent MRI findings were bilateral, multifocal hyperintense signal abnormalities on fluid-attenuated

**Fig. 5 | Individual CSF and plasma analytes discriminating different groups.** Box plot representations (center line at the median, upper bound at 75th percentile, lower bound at 25th percentile) of marginalized normal protein expression (NPX) of individual analytes significantly discriminating selected groups with whiskers at minimum and maximum values. Each dot represents one participant **a** Plasma, increasing NPX from *class I (n = 18) to III (n = 15), and higher than in controls (n = 50):* IL-6 (class III vs. I: $p = 0.007$, class III vs. infl. ctrl: $p = 0.001$), IL-8 (class III vs. I: $p = 0.003$, class III vs. infl. ctrl: $p = 0.0002$), HGF (class III vs. I: $p = 0.04$, class III vs. infl. ctrl: $p = 0.0007$), VEGFA (class III vs. I: $p = 0.01$, class III vs. infl. ctrl: $p = 0.0005$), EN-RAGE (class III vs. I: $p = 0.003$, class III vs. infl. ctrl: $p = 0.0002$), TNFRSF12A (class III vs. I: $p = 0.006$, class III vs. infl. ctrl: $p = 0.002$), PD-L1 (class III vs. I: $p = 0.04$, class III vs. infl. ctrl: $p = 0.002$), CCL23 (class III vs. infl. ctrl: $p = 0.01$), EZR (class III vs. infl. ctrl: $p = 0.002$), TNFRSF11B (class III vs. infl. ctrl: $p = 0.049$). **b** Plasma, decreasing NPX from *class I (n = 18) to III (n = 15), higher in controls (n = 50) than in COVID-19 patients (n = 40):* BMP-4, CLEC10A, CNTN5, GDF-8, NTRK2, GDNFRalpha, ROBO2. **c** Plasma, class-independent, *higher NPX in COVID-19 (n = 35) than in controls (n = 50):* 4E-BP1 (class III vs. infl. ctrl: $p = 0.007$). **d** Plasma, *higher NPX in COVID-19 (n = 40) than in controls (n = 50), decreasing from class I (n = 18) to III (n = 15):* HAGH (class III vs. I: $p = 0.008$). **e** Cerebrospinal fluid (CSF), *increasing NPX from class I (n = 16) to III (n = 14):* IL-8 (class III vs. I: $p = 0.012$), MSR1 (class III vs. I: $p = 0.016$), 4E-BP1 (class III vs. I: $p = 0.02$), CD200R1 (class III vs. I: $p = 0.04$), TNFRSF12A (class III vs. I: $p = 0.008$), EZR (class III vs. I: $p = 0.01$). **f** CSF, *increasing NPX from class I (n = 16) to III (n = 14), and higher in class III than in inflammatory controls (n = 25):* TNFRSF11B (class III vs. I: $p = 0.04$, class III vs. infl. ctrl: $p = 0.02$). Source data are provided as a Source Data file. Statistics (**a–f**): statistical significance was calculated using two-sided Mann–Whitney-U test and $p$ values were adjusted using Benjamini-Hochberg (BH)-procedure ($p$: *<0.05; **<0.01; ***<0.001, if not otherwise indicated: not significant).

inversion recovery (FLAIR)/T2-weighted (T2w) imaging ($n = 18$, 56.3%; class I: $n = 5$, 33.4%; class II: $n = 5$, 83.4%; class III: $n = 11$, 72.7%). These signal abnormalities were predominantly located in the periventricular region (13 patients, 40.6%) and the semioval center (16 patients, 50%). Additional FLAIR/T2w signal abnormalities were observable in the *corpus callosum* (9 patients, 28.1%) and in the brain stem (7 patients, 21.9%). In one class III patient, bilateral thalamic signal hyperintensities were present on T2w imaging (Fig. 7c). Furthermore, diffusion-weighted imaging (DWI) changes were present in 4 patients (12.5%): 1 class I/II, and 2 class III patients. Black blood and/or time of flight (TOF) imaging was acquired in 4 patients, in 2 of which (both from class III) focal vessel wall enhancement was visible, indicative of cerebral vasculitis (Fig. 7c). No signal changes were detected in the olfactory bulb. In 3 CT-scanned class III patients, we found 1 infratentorial or supratentorial infarction, 1 thrombosis of the sigmoid sinus with intracerebral hemorrhage and 1 bifrontal subarachnoid hemorrhage (Fig. 7c).

Within our study cohort, we identified 6 patients with pre-existing MRI scans (3 class I, 3 class III patients), whereas one class III patient had a brain MRI followed by CT 1 week later during COVID-19. 3/6 patients had imaging alterations compared to their pre-COVID-19 MRIs (Fig. 7d–f). In one class I patient, we detected hyperintense signal alterations in the left cerebellar hemisphere (Fig. 7d) and in the left frontal juxtacortical region on DWI (Fig. 7e), suggesting small territory acute diffusion restriction. The other two patients' brain scans demonstrated nonspecific white matter hyperintensities on FLAIR/T2w imaging (Fig. 7f). In the class III patient with MRI and subsequent CT scan, the MRI scan was unremarkable, whereas the CT depicted right sided cerebellar infarction (Fig. 7c).

## Lower GMVs in olfactory pathway structures in Neuro-COVID patients are negatively correlated to inflammatory CSF parameters

To identify implications of the deranged protein landscape to brain integrity, we assessed GMVs in different brain areas and their association to routine and experimental CSF and plasma parameters. There were no significant differences between the Neuro-COVID group and the imaging control group in age, sex and global brain variables (Supplementary Table 5).

After false discovery rate (FDR) correction, we identified 16 specific brain regions that were negatively correlated to the CSF leukocyte count, protein levels and the CSF/plasma albumin ratio ($p < 0.05$) (Fig. 7g–j, Supplementary Table 6). Thereof, 81% corresponded to the olfactory and gustatory cortex's telencephalic connections, including the amygdala, entorhinal cortex, basal ganglia, cingulate gyrus and orbitofrontal areas (Fig. 7g–j).

Alternatively, we investigated the correlation of CSF and plasma mediators with decreased regional GMVs. For instance, high plasma levels of BMP-4 and GDF-8 were associated with preserved regional GMVs (Supplementary Fig. 7), whereas PD-L1 and HGF were associated with decreased GMVs in specific brain regions (Supplementary Fig. 8).

Additional proteins associated with GMV alterations are depicted in Supplementary Fig. 9. However, none of the $p$ values were significant after BH procedure, highlighting the need for larger sample sizes and targeted analysis of these plasma proteins.

## Long-COVID is more prevalent in severe Neuro-COVID patients and associated with specific CSF and plasma parameters

We further investigated the potential of CSF and plasma proteins to predict long-COVID in a 13 months follow-up. Long-COVID was defined according to the WHO consensus definition[38]. Deceased patients were not taken into account for the long-COVID prediction analysis.

Detailed results of the 13 months follow-up are described in Table 1. Due to the high mortality rate and patients lost to follow-up within class III (5/15 patients deceased, 7 lost to follow-up), we performed only 3 follow-ups in this group. Out of the deceased patients, three died during their hospital stay and 2 during the follow-up period. Class II and III patients were more often affected by long-COVID compared to class I. In class I, 11/18 patients recovered without long-term deficits, whereas 6/7 class II and 3/3 patients in class III had long-COVID. Furthermore, the death of a class I patient 3 months after COVID-19 was caused by known end-stage heart failure.

Using an AUC cut-off of >0.75, high single plasma CLM-6 (CD300c), MCP-3[39,40], and low RGMA[41] levels were predictive for long-COVID (Fig. 8a). Furthermore, confusion matrix analysis of plasma proteins revealed high predictive power of a signature of EZR, RGMA, FcRL2 and ST1A1 for long-COVID forecast[28,29,41–43] (Fig. 8a).

Within CSF, low levels of TRANCE (RANKL), as well as high TNFRSF9 and IFN-γ levels were the best single protein predictors for long-COVID[1,2,44–46] (Fig. 8b). Confusion matrix analysis of CSF analytes revealed high predictive power of a pattern composed of TRAIL, IFN-γ and TNFRSF9[12,44,47–49] (Fig. 8b).

## Discussion

We identified Neuro-COVID-specific CSF and plasma alterations, providing insights into pathomechanisms underlying COVID-19-related neurological sequelae. Compared to previous analyses, we studied the associations of peripheral inflammation, neuroinflammation and NS multidimensionally within prospectively stratified Neuro-COVID classes and COVID-19 severity degrees.

### Class III patients display a non-inflammatory CSF profile, signs of blood-brain barrier disruption and a polyclonal antibody response in the CSF, potentially caused by systemic immune dysregulation

In line with other studies, we found elevated CSF glucose and lactate levels in class III patients. Indeed, patients suffering from diabetes type II were more prevalent in class III (Table 1). Nonetheless, the CSF/plasma glucose ratio was significantly higher in class III patients compared to class II patients, pointing towards not only diabetes-related causes of elevated CSF glucose levels. The elevated lactate levels in

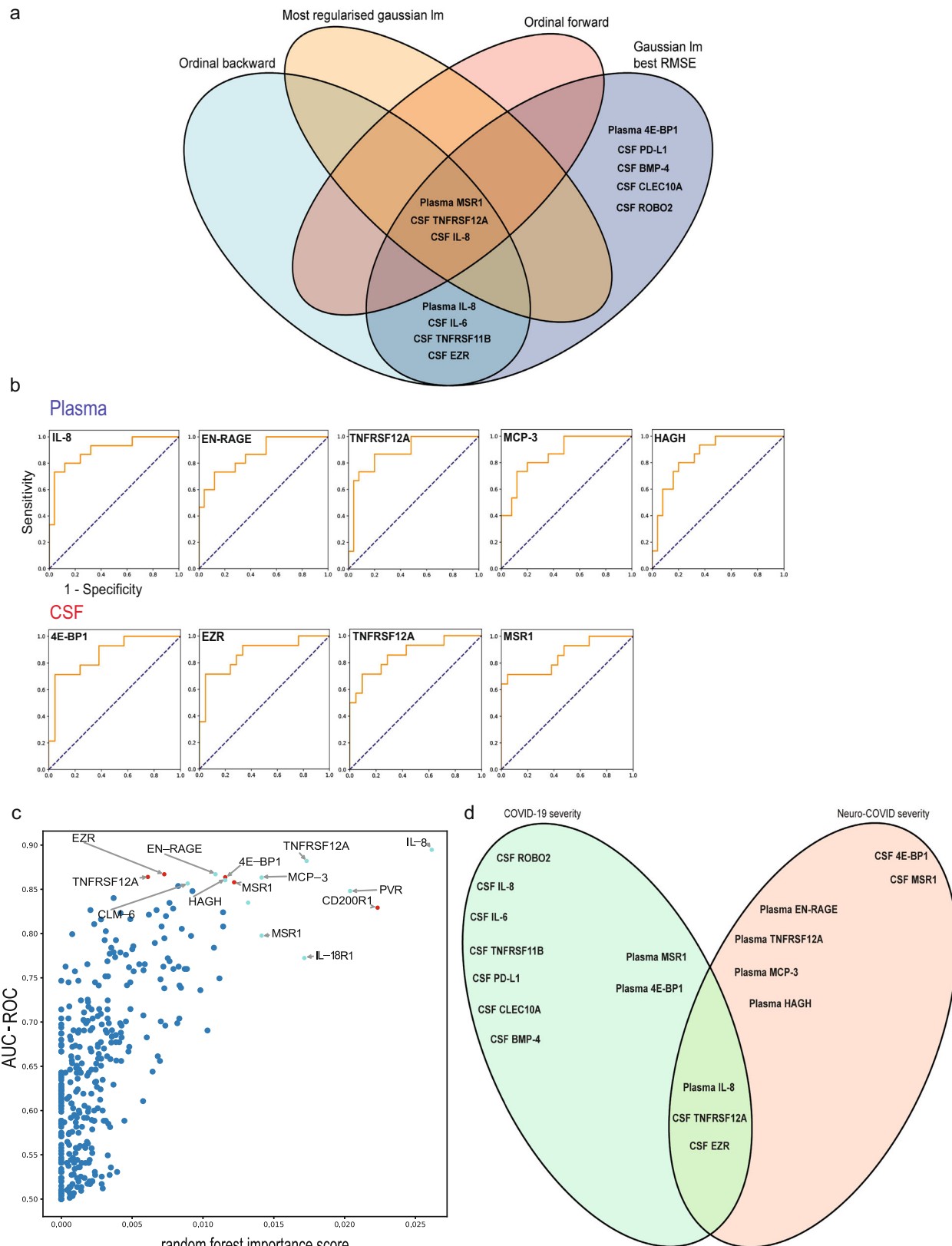

class III potentially hint at cerebral hypoxia. For instance, class III patients had putative COVID-19-induced stroke or intracerebral hemorrhage, which may explain this finding.

Notably, we identified a class III-specific humoral CSF immune response encompassing enrichment of (total) IgG/IgA against self-(dsDNA) and non-self (BSA) antigens. This finding is corroborated by distinct plasma B-cell clusters and CSF antibody reactivity profiles previously reported[12]. Certainly, antibody production predominantly took place in the plasma, pointing to an ingress of peripherally activated B cells and antibodies. In line with recent findings identifying new-onset autoantibodies in patients with COVID-19, our identification of elevated anti-dsDNA IgG, which are associated with cardiovascular symptoms in

**Fig. 6 | Specific CSF and plasma analytes correlate with COVID-19 severity and have predictive value for class III development. a** Venn diagram representing cerebrospinal fluid (CSF) and plasma mediators associated with COVID-19 severity assessed with the WHO progression scale. The association of protein sets and COVID-19 severity was assessed using a complement of four models (ordinal backward, ordinal-forward, best linear, most-regularized ordinal). Results of each model were finally cross-referenced to provide robust data sets of mediators associated with severe COVID-19. Plasma MSR1 and CSF TNFRSF12A and IL-8 represent the most robust set of proteins associated with COVID-19 severity (high association in each model used). Plasma IL-8 and IL-6, TNFRSF11B and EZR CSF levels depicted strong association using two models, whereas plasma 4E-BP1 and CSF PD-L1, BMP-4, CLEC10A and ROBO2 were strongly associated with COVID-19 severity in one model only. **b** ROC-AUC analysis of class I and II vs. class III. Five

predictive plasma markers, including IL-8, EN-RAGE, TNFRSF12A, MCP-3, and 4 CSF markers, including 4E-BP1, EZR, TNFRSF12A, MSR1, emerged for the prediction of class III development. The Y-axis represents the sensitivity, the X-axis represents the 1-specificity (represented for IL-8, plasma). The names of relevant proteins in the study are compiled in Supplementary Data 4. **c** Plot representing the ROC-AUC values on the Y-axis vs. the random forest importance score on the X-axis. Relative importance of each single protein is represented by a high random forest importance score. *Red points: plasma proteins, light blue points: CSF proteins* **d** Venn diagram representing CSF and plasma mediators associated with COVID-19 severity assessed with the WHO progression scale (COVID-19 severity) and mediators with high predictive value for class III development (Neuro-COVID severity). Plasma IL-8, CSF TNFRSF12A and CSF EZR depicted a high predictive value for severe COVID-19 and also class III development. lm linear model, RMSE root mean squared error.

systemic lupus, may provide a potential pathophysiological rationale for cardiovascular risk factors in severe COVID-19[11,12,35,50–52]. Indeed, two out of four class lll patients with vascular complications had increased levels of anti-dsDNA IgG in the CSF.

Our observation of elevated anti-gut microbial IgA antibodies supports the evidence for gut barrier dysfunction in severe COVID-19 that may necessitate containment of gut microbiota translocated to the circulation and possibly the CSF[53–55]. In that regard, underlying conditions for microbiota dysbiosis, such as increased age, hypertension and diabetes have been observed in our class III cohort (Table 1). Alternatively, trafficking of commensal-reactive regulatory B cells to sites of neuroinflammation as recently described may underlie these findings[56,57]. Our observations shed new light on mucosal barrier disruption as a modulator of the peripheral host immune response. While we cannot exclude that the differences in class III are due to pre-existing antibody profiles, the class-dependent increase in polyclonal antibody responses argues for a COVID-19-related pathophysiology and is corroborated by recent findings identifying new-onset auto-antibodies in patients with COVID-19[11].

Intriguingly, total CSF antibody levels were significantly higher in class III compared to class I/II patients (Supplementary Fig. 1d), representing a severity-dependent, compartmentalized B-cell response, likely induced by BBB disruption.

Based on the results of our BCR sequencing, patients producing more B-cell clones in the periphery may develop milder disease for two reasons: First, these patients produced higher levels of total IgG, which might be protective, and second, the expanded clones produced more specific antibodies in comparison to severely affected patients. In class III patients, the antibody response might be more diverse due to enhanced polyclonality, reflecting an unspecific and dysregulated immune activation, as seen in autoimmune diseases[12,58].

## Cytokine-induced blood-brain barrier disruption might lead to heightened polyreactive antibody ingress to the CNS

Higher CSF protein, albumin, CSF/plasma ratio and IgM, IgG and IgA levels, and intrathecal detection of peripherally produced SARS-CoV-2 S-antibodies underscored BBB impairment in class III patients. Of note, SARS-CoV-2 S-antibody levels increase with a decreasing viral load[59], explaining our low detection rate.

In line with prior research[16,17], we observed a class-incremental cytokine storm in plasma, but less prominent in the CSF. Intriguingly, class III patients displayed a unique CSF protein pattern highlighting BBB disruption, microglia regulation and neuronal tissue damage. TNFRSF12A displayed a high CSF-plasma correlation and predictive value for class III development, rendering it a predictive biomarker for severe Neuro-COVID given its involvement in BBB disruption during CNS immune cell recruitment[60,61]. Further, IL-8, VEGFA and EN-RAGE promoted class III inherent BBB impairment, reinforcing ingress of polyreactive antibodies into the CSF[22,27,62–64].

Notably, several of the proteins with higher CSF and plasma levels in class III patients were associated with COVID-19 severity[14]. Plasma 4E-

BP1, MSR1, IL-8 and IL-6 as well as CSF TNFRSF12A, TNFRSF11B, CLEC10A, PD-L1 and EZR were associated with COVID-19 severity *and* Neuro-COVID, highlighting the role of the previously described cytokine storm, the innate immune system, particularly microglia overactivation, and a dysfunctional BBB in progressive COVID-19[18,19,22,27,45,46,65].

## Microglia overstimulation affects cerebral integrity in COVID-19

Exploiting microglia, neuronal markers and neuroimaging, we investigated consequences of COVID-19-induced BBB impairment on cerebral integrity. TNFRSF11B, a decoy receptor for TRANCE (RANKL), was the sole CSF discriminant between class III and CNS inflammatory controls, leading to microglia overstimulation[45]. Importantly, we detected concurring elevated TRANCE (RANKL) CSF/plasma ratios in class III, which underscores the relevance of increased TNFRSF11B levels. Targeting TNFRSF11B, e.g., by TRANCE (RANKL) mimics[46], could attenuate microglia activity in Neuro-COVID. Another relay of propagating the peripheral inflammation to the brain is represented by elevated MSR1 levels[66] followed by elevated CD200R1 levels in class III. Altogether, this suggests microglial activation and its possible consequences in severe Neuro-COVID.

## Structural brain imaging alterations dominate in class III patients

Brain imaging revealed findings in class III patients, while most class I/II patients lacked evidence of profound alterations. Findings in class I/II patients mainly consisted of white matter FLAIR/T2w abnormalities, which have a broad spectrum of differential diagnoses, and are common in the elderly[52,67,68]. Therefore, we cannot deduct a causality of COVID-19 and unspecific brain imaging changes, which necessitates pre-COVID-19 scans. However, 3/6 patients with pre-COVID scans from our cohort had novel alterations, pointing towards a possible association with the disease. Further, COVID-19 has been postulated to cause endothelial dysfunction, microangiopathy and a prothrombotic state, possibly explaining the impressive pathologies observed in our class III patients[52,69]. Nonetheless, the underlying pathomechanisms resulting in brain imaging abnormalities and the direct implication of SARS-CoV-2 remain unsolved. Further, particularly longitudinally designed studies assessing pre- and post-COVID-19 brain scans could help identify potential pathomechanisms related to SARS-CoV-2.

## Standard CSF parameters associate with reduced olfactory GMVs in COVID-19 patients

Recently, a large scale, longitudinal volumetric brain imaging study of COVID-19 patients reported results in line with our findings[8]. However, the authors did not report associations of volumetric brain alterations with particular CSF and plasma biomarkers. In our cohort, GMVs of olfactory pathway regions were negatively correlated with the CSF/plasma albumin ratio, CSF leukocytes and protein levels in COVID-19 patients. The pattern of decreased GMVs, particularly in olfactory pathway structures, is consistent with reports of decreased glucose

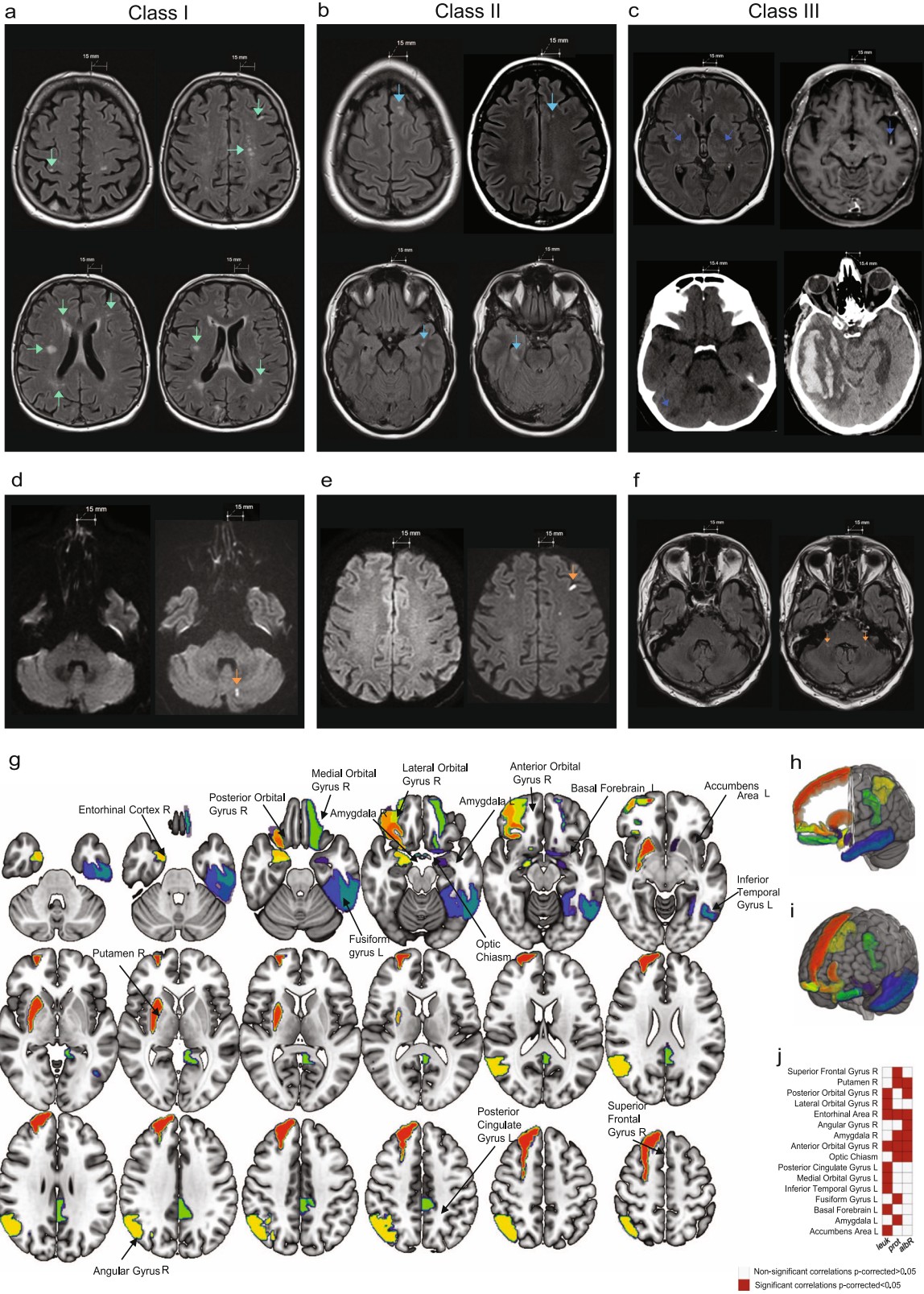

metabolism in fronto-parietal and temporal regions[8]. Maybe, the higher CSF glucose levels and CSF/plasma glucose ratios in severe Neuro-COVID can be explained by a lower glucose turnover of the brain, particularly in regions where GMVs were negatively associated with inflammatory CSF parameters. Likewise, some of these regions with decreased GMVs have overlapping olfactory- and memory-related functions, which may explain long-term neurological sequelae, such as memory or concentration problems after COVID-19[8].

**COVID-19 severity associated biomarkers could impact GMVs in particular brain regions**

In line with previous research, high PD-L1 and HGF plasma levels were associated with COVID-19 severity[17,30]. Potentially, PD-L1 blockade would counteract the previously described immune dysregulation[30]. Conversely, HGF, reported to mediate tissue-regenerative responses in COVID-19-induced lung damage[17], might serve as a counter-regulatory factor promoting neuroregeneration

**Fig. 7 | Routine brain imaging, regional GMVs and association with inflammatory CSF parameters. a–c** Conventional brain MRI and CT scans depicting exemplary imaging findings. Scale bar 15 mm (MRI) and 15.4 mm (CT). **a** Class I: Axial FLAIR images of the same class I patient show multifocal hyperintensities in the right precentral gyrus *(top left)*, semioval center, left frontal cortex *(top right)*, deep white matter, periventricular region *(bottom left)*, right temporal lobe, left parietal white matter *(bottom right)*. **b** Class II: Axial FLAIR images of a class II patient depict multifocal hyperintensities in the left frontal superior gyrus *(top left)*, white matter of left frontal lobe *(top right)*, left parahippocampal white matter *(bottom left)*, right mesial temporal region *(bottom right)*. **c** Class III: Axial FLAIR image shows bilateral thalamic hyperintensities *(top left)*. Axial T1-weighted image depicts left middle cerebral artery (M2-segment) enhancement in the insular cistern *(top right)*. Coronal CT scans demonstrate right cerebellar infarction *(bottom left)* and right temporo-occipital intracerebral hemorrhage *(bottom right)* (secondary to thrombosis of the right sigmoid sinus). **d–f** Conventional pre-COVID-19 MRIs *(left)* and MRIs during COVID-19 *(right)*. **d, e** Axial DWI of the same class I patient depicts hyperintense signal alterations during COVID-19 *(right sided MRI scans, orange*

*arrows)* in the left cerebellar hemisphere (**d**) and frontal juxtacortical region (**e**). **f** Axial FLAIR imaging demonstrates bilateral hyperintense signal alterations in the cerebellar peduncles *(right sided MRI scan, orange arrows)* of a class III patient. **g–i** Map (**g**) and 3D view (**h, i**) of the 16 brain regions with significant correlation values of gray matter volume (GMV) and clinical variables in the Neuro-COVID group after multiple comparison correction (FDR). These regions are represented in different colors on a T1-weighted template. **j** shows a matrix representing the association significance (significant p-corrected <0.05 in red squares). Associations between regional volumes and clinical measures were assessed using partial correlation, allowing to calculate the linear partial correlation between variables of interest adjusting for different covariates (age, sex, age*sex interaction, MRI magnetic field strength, total intracranial volume (TIV)). Statistical analysis was performed using the JASP software (https://jasp-stats.org/). MRIcroGL software was used to generate this figure (https://www.nitrc.org/projects/mricrogl). Source data of (**g–j**) are provided as a Source Data file. L left, R right, leuk leukocytes, prot protein, albR Albumin CSF-plasma ratio.

upon neuronal tissue damage. Also, high PD-L1 and HGF plasma levels were associated with decreased GMVs in particular brain regions in other contexts[17,30], whereas BMP-4 and GDF-8 were associated with preserved volumes[32,70,71]. However, the results were not significant, which could be explained by the low number of patients providing both CSF/plasma and required MRI sequences for brain volumetric analysis. Associating PD-L1 and HGF levels with volumetric brain changes in a larger patient population could provide new actionable targets to prevent short- and long-term neurological sequelae. Recently, work from ref. 72 pointed at the impact of CCL-11 on neuronal damage and microglia activation in COVID-19, corroborating the impact of peripheral cytokines on neuronal and microglial pathology in COVID-19.

### Long-COVID is predicted by a peripheral and CNS innate immune dysregulation

Our long-term follow-up suggests that class II and III patients continued to be more frequently affected by long-COVID compared to class I patients. Furthermore, we identified CSF levels of pro-inflammatory proteins (TNFRSF9, IFN-γ) and lacking anti-inflammatory mediators (TRANCE(RANKL), TRAIL) to be predictive for long-COVID, whereas plasma CLM-6, MCP-3 and ST1A1 revealed potential for long-COVID forecast[39,40]. The association of particular pro-inflammatory cytokines and long-COVID, amongst others IFN-γ, have been previously described by Phetsouphanh et al.[73] and is in line with our finding of elevated CSF IFN-γ levels in long-COVID.

MCP-3 is crucial for efficient macrophage infiltration into the CNS[40,74], disclosing a dominant role of the innate immune system in COVID-19-related long-term NS. Furthermore, CLM-6 upregulation reflects a pronounced monocyte activation, underscoring the connection of innate immune effectors with long-COVID development[39]. Strikingly, ST1A1 was upregulated in an experimental autoimmune encephalitis model[42], pointing at autoimmune mechanisms in COVID-19, and aligning with autoreactive antibodies in severe COVID-19 and Neuro-COVID. Recently, work from Su et al. pointed at the association of atypical memory B cells, exhibiting lower levels of somatic hypermutation and enhanced BCR and IFN signaling in long-COVID, sharing pathomechanisms with systemic lupus. This observation is in line with our finding of elevated mediators of autoimmunity and emphasizes the involvement of autoimmune mechanisms in long-COVID development.

Certainly, our analysis has limitations. Although prospectively designed, we do not provide longitudinal follow-up data of assessed parameters. However, we provide a 13 months questionnaire-based follow-up confirming long-term neurological sequelae and higher mortality rates in class III.

Moreover, we recruited a relatively low number of class II patients, precluding us from characterizing this class to the same extent as class I and III.

Only unvaccinated patients were included since we recruited before the roll-out of COVID-19 vaccinations. Studies on the impact of vaccinations on reported findings might be of clinical relevance.

We provide a multiparametric framework of Neuro-COVID severity classifiers. The main determinants of severe Neuro-COVID are: (1) peripherally induced cytokine derangements, followed by (2) impaired BBB with ingressing polyreactive autoantibodies, (3) microglia reactivity and neuronal damage resulting in (4) potential GMV loss, possibly explaining short- and long-term COVID-19-related neurological impairment (Fig. 9). Collectively, these data identified several possible targets which should be further investigated to potentially prevent COVID-19-related long- and short-term neurological sequelae.

## Methods
### Ethics oversight, patient recruitment, follow-up and reporting of data
This research project complies with all relevant ethical regulations. The study and all uses of human material was approved by the Ethics Committee of Northwestern and Central Switzerland (clinicaltrials.gov NCT04472013, IRB approval EKNZ 2020-01503). The trial protocol can be found in the Supplementary Information. Patients (*n* = 40) were recruited during a period from August 2020 to April 2021 at two sites, the University Hospitals Basel and Zurich. Patients were recruited at the COVID-19 test center, the hospital ward or at the intensive care unit. For each participant, written informed consent was obtained. If the participant was not able to provide written informed consent, written informed consent was obtained by their relatives. For their additional hospital visit, patients recruited at the test center were paid 200 Swiss Francs.

Inclusion criteria were age ≥18 years and a real-time quantitative PCR (qRT-PCR)-positive SARS-CoV-2 infection. The only applied exclusion criteria was pregnancy. A 13 months patient reported outcome follow-up was performed using the modified COVID-19 Yorkshire Rehabilitation Screening (C19-YRS).

We confirm that our study fully complies with the STROBE statement and the STARD guidelines.

### CSF and plasma sampling
All uses of human material have been approved by the Ethics Committee of Northwestern and Central Switzerland (clinicaltrials.gov NCT04472013, IRB approval EKNZ 2020-01503). Out of 40 COVID-19 patients (mean [SD] age, 54 [20] years; 17 women (42%)

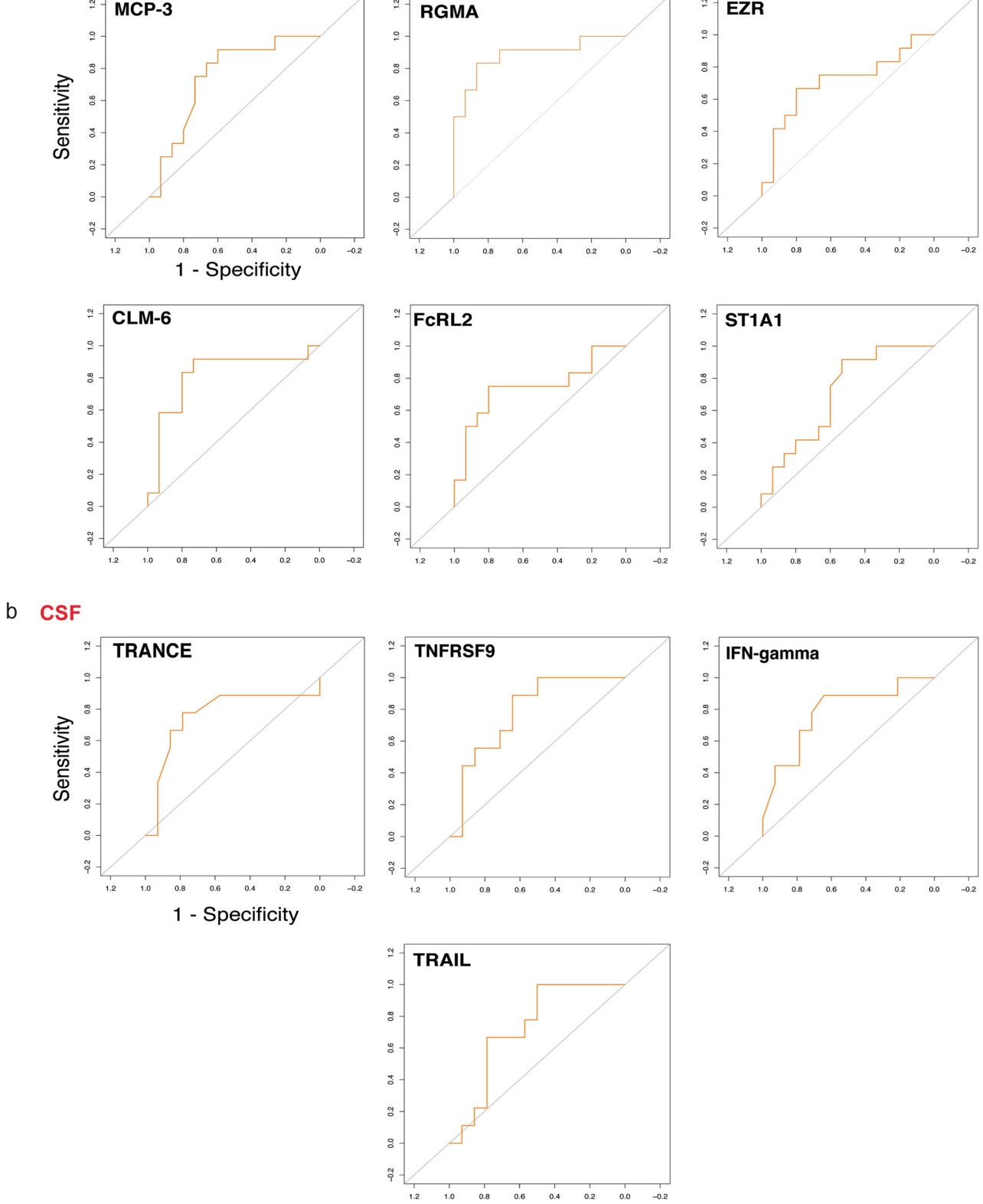

**Fig. 8 | Specific CSF and plasma mediators have high predictive value to fore-cast long-COVID.** ROC-AUC analysis of CSF and plasma parameters and long-COVID. The Y-axis represents the sensitivity, the X-axis represents the 1-Specificity (represented for plasma MCP-3). **a** Assuming an AUC cut-off of >0.75 for single mediators, high MCP-3 and CLM-6, as well as low RGMA plasma levels were pre-dictive for long-COVID. Confusion matrix analysis of predictive plasma proteins revealed high predictive value of a plasma mediator signature consisting of RGMA, EZR, FcRL2 and ST1A1. **b** Assuming an AUC cut-off of >0.75, three single CSF pro-teins emerged for the prediction of long-COVID development. Low CSF levels of TRANCE, as well as high TNFRSF9 and IFN-γ levels were the best predictors. Con-fusion matrix of CSF proteins revealed high predictive power of a mediator pattern composed of TNFRSF9, IFN-γ and TRAIL.

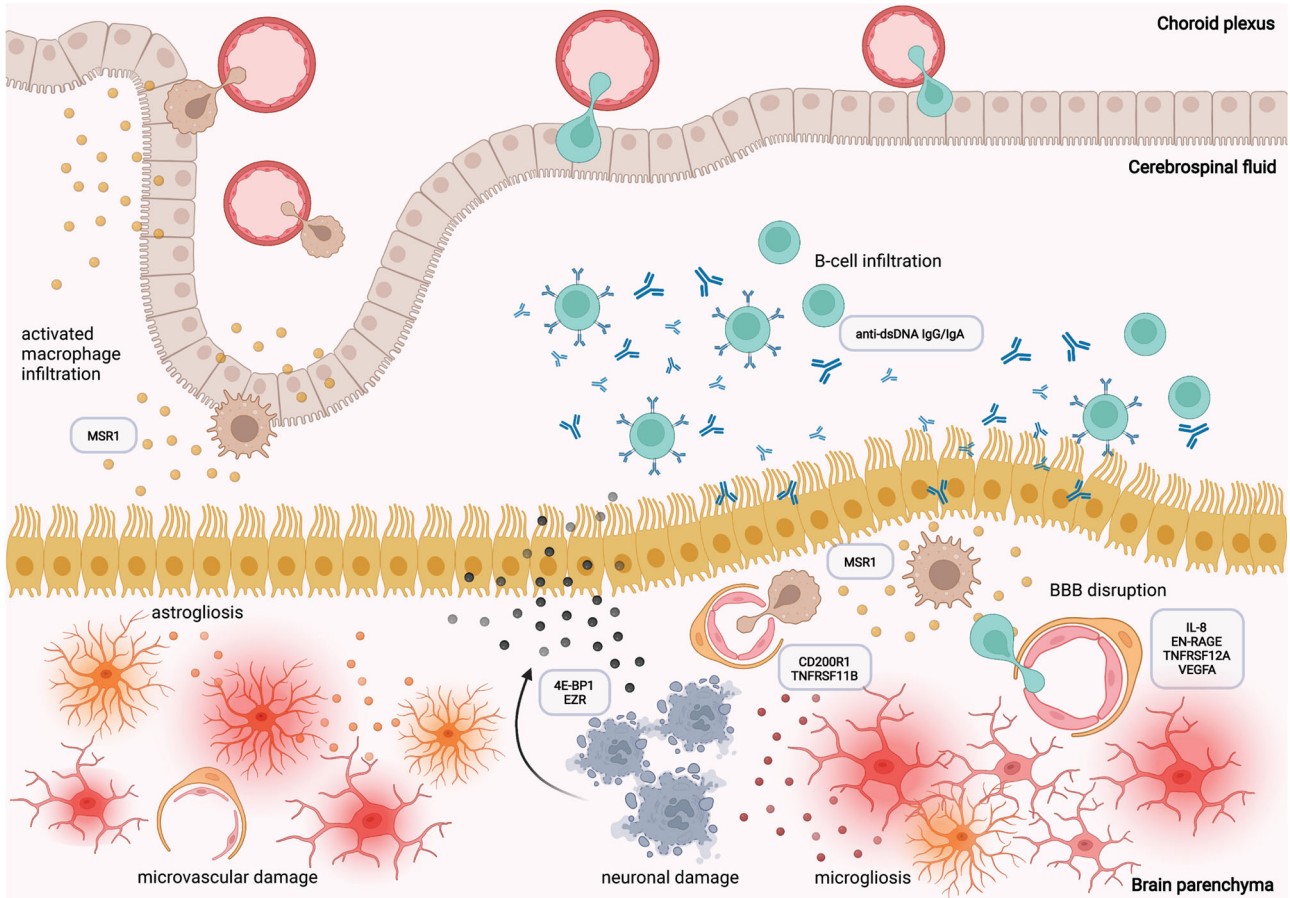

**Fig. 9 | Overview of proposed pathomechanisms leading to Neuro-COVID.** The proposed main determinants of severe Neuro-COVID are: (1) peripherally induced cytokine derangements, followed by (2) impaired BBB with ingressing polyreactive autoantibodies, resulting in (3) microglia reactivity and neuronal damage. Created with Biorender.com.

patients, 35 donated paired blood and CSF samples, whereas 5 participants donated only blood samples. Lumbar puncture (LP) and blood withdrawal were performed concomitantly, on an average latency period of 4 days after the first positive SARS-CoV-2 qRT-PCR test result. LP was performed under sterile conditions using a 20 gauge needle under local anesthesia on lumbar midline levels L4/5. Patients were monitored for positional headache or signs of CSF leakage for 24 h after puncture. Fresh CSF and EDTA-treated blood samples were processed into CSF supernatant and plasma. CSF samples were processed within 30 min post collection. After centrifugation at $1000 \times g$ for 10 min, cell-free supernatant was removed, aliquoted and stored at −80 °C. Whole blood was first centrifuged at $2000 \times g$ for 10 min to separate plasma and blood cells. The isolated plasma was then centrifuged at $1000 \times g$ for 10 min to remove residual blood cells, aliquoted and stored at −80 °C and subsequently liquid N2. Retrospectively biobanked, age- and sex-matched paired CSF and plasma samples from patients with non-MS inflammatory neurologic disorders ($n = 25$; mean [SD] age, 54 [19] years; 12 women [48%]) and healthy donors ($n = 25$; mean [SD] age, 52 [18] years; 12 women [48%]) (Supplementary Table 1) served as controls and were obtained from J.K. and J.O., Neurology Department, University Hospital Basel.

**Antibody assays.** The COVID-19 cohort (mean [SD] age, 54 [20] years; 17 women (42%) and control samples (CNS inflammatory controls: $n = 25$; mean [SD] age, 54 [19] years; 12 women [48%], and healthy donors: $n = 25$; mean [SD] age, 52 [18] years; 12 women [48%]) were always analyzed together in the same batch. Laboratory personnel were unable to make a difference between COVID-19 patient samples and control samples.

## Quantification of total immunoglobulins and SARS-CoV-2 spike antibodies
Immunoglobulin (Ig) levels and anti-SARS-CoV-2 spike (S) protein IgG in plasma and CSF were quantified using nephelometric and ELISA assays and AI indices were calculated as part of the clinical routine diagnostic.

## Anti-MOG and anti-NF155 antibody assays
Paired plasma and CSF supernatant samples from COVID-19 patients, healthy and CNS inflammatory controls were examined for IgG reactivities against conformational human myelin oligodendrocyte glycoprotein (hMOG) and neurofascin-155 (NF155)[50,75–77] using cell-based assays as previously described. In brief, stably transfected TE cells expressing full-length MOG, NF155 or the respective empty vector control were incubated with plasma (1:100) or CSF (1:5) and antibody binding was detected using secondary anti-human IgG-PE (Jackson). Humanized MOG- (h818C5; 0.2 μg/mL) or NF155-specific (A12/18.1; 0.6 μg/mL) monoclonal antibodies were included as positive controls, respectively. Live cells were measured on a CytoFLEX flow cytometer and data analysis was performed in FlowJo (FlowJo 10.6.2, Becton Dickinson and Company). The ratio of the geometric mean fluorescence intensity (MFI) of the transfected cell line divided by the MFI of the control cell line was calculated. The cut-off was set to 3 standard deviations above the mean of a healthy control cohort.

## Commensal bacteria and polyreactivity ELISA

Human gut commensal bacteria, comprising 33 commensal bacteria strains (RePOOPulate)[56], double-stranded DNA (UltraPure Salmon Sperm DNA, Thermo Fisher) and bovine serum albumin (BSA, Sigma-Aldrich) were coated on a MaxiSorp ELISA plate (Nunc) in PBS in triplicates and incubated overnight at 4 °C as recently reported[56]. Plates were washed and blocked with 3% BSA in PBS for 2 h at RT before incubation with plasma (1:100) or CSF (1:5) for 1 h. After incubation with anti-human IgG or IgA horseradish peroxidase (Jackson ImmunoResearch) for 1 h, the assay was developed with TMB peroxidase substrate (Seracare). A polyclonal polyreactive IgG antibody (ED-38; undiluted transfection supernatant) was used as a positive assay control[67]. Triplicates with a coefficient of variation (CV) greater than 15% were corrected for by excluding one value. Corrected duplicates with a CV above 15% were excluded from the analysis ($n = 4$). Negative control signals (secondary antibodies only) were subtracted in a plate-specific manner.

## RNA extraction, library preparation and BCR sequencing

Total RNA was extracted from peripheral blood mononuclear cells (PBMC) of exemplary class I ($n = 4$; mean [SD] age, 64 [10] years; 2 women [50%]) and class III ($n = 4$; mean [SD] age, 64 [13] years; 1 woman [25%]) patients using the AllPrep DNA/RNA Mini Kit (Qiagen, 80204) following the vendor's instructions. RNA concentration was measured with Qubit RNA HS Assay Kit (ThermoFisher, Q32852). cDNA was synthesized from 25 ng total RNA by Ion Torrent NGS Reverse Transcription Kit (ThermoFisher, A45003). Library preparation was done with Oncomine BCR IGH SR RNA Assay (ThermoFisher, A45484). Amplified and barcode ligated libraries were purified with AMPure XP Reagent (Beckman Coulter, A63880) and quantified with Ion Universal Library Quantitation Kit (ThermoFisher, A26217). Library pool was prepared by combining equal volumes of libraries at 50 pmol/L concentration and loaded into Ion 550™ Chip (ThermoFisher, A34537). The libraries were sequenced with Ion GeneStudio S5 Prime Sequencer, ThermoFisher). All clones used for BCR sequencing are depicted in Supplementary Data 8.

## BCR sequencing data analysis

The BCR sequencing analysis was done by Ion Reporter Software Version 5.18 (ThermoFisher). Global immune repertoire metrics such as B-cell clone number (richness), Shannon diversity index and evenness (normalized Shannon's diversity index) were calculated to describe the diversity of the B-cell clones in the blood. Shannon diversity index calculation takes into account the total clone number (richness) and evenness of the clones. Evenness measures the relative clonal abundancy and it has a value between 0 and 1. When the evenness approaches 0, it shows an unbalanced clone distribution with high frequency clones in the population, or vice versa evenness close to 1 means an even distribution of clones. CDR3 nucleotide length of each clone was calculated and CDR3 length distribution was plotted with R Studio Version 4.1.2. ggplot 2.

## Multiplexed secreted protein assays from CSF and plasma.

A total of 192 analytes, including chemokines, soluble cell membrane proteins and cytokines, were measured in 85 paired plasma and CSF supernatant samples and additional 5 COVID-19 plasma samples, acquired from COVID-19 patients ($n = 40$; mean [SD] age, 54 [20] years; 17 women (42%), healthy controls ($n = 25$; mean [SD] age, 52 [18] years; 12 women [48%]) and patients with non-COVID-19 non-MS inflammatory neurological disorders ($n = 25$; mean [SD] age, 54 [19] years; 12 women [48%]). The Olink 96 target neurology (https://www.olink.com/products-services/target/neurology-panel/) and Olink 96 target inflammation (https://www.olink.com/products-services/target/inflammation/) panels were used. The measurements were performed by the Olink Analysis Service at Olink laboratories

(SIAF, Davos, Switzerland). The assay used oligonucleotide-labeled antibody pairs allowing for pair-wise binding to target proteins. Briefly, when antibody pairs bound target antigens, corresponding oligonucleotides hybridized and were extended by polymerases and formed a unique barcode, allowing the quantification of protein analytes by high-throughput RT-PCR. Data are presented as normalized protein expression values, Olink Proteomics' arbitrary unit on a $\log_2$ scale. Missing data were associated with a lower median expression. They were imputed as either half the molecule detection threshold or such that the sum of all imputed values for a molecule is 0.1 of the sum of the molecule's expressions, whichever was the smallest.

**Analysis of multiplexed protein expression data.** The COVID-19 cohort and control samples were always analyzed together in the same batch. Laboratory personnel were unable to make a difference between COVID-19 patient samples and control samples.

## Handling of missing data/imputation

All values below the molecule-specific detection thresholds provided with the experimental results were treated as missing data which were imputed as either half the detection threshold or such that the sum of all imputed values on the column is 0.1 of the column values sum, whichever was the smallest.

## Marginalization of age and sex

To account for discrepancies in the distribution of age and sex between the different groups, we used a linear model on the normalized protein expression (NPX) values, and marginalized individual observations for the median age and the female sex. Marginalization reports for each molecule are documented and available in the Supplementary Material.

## Single cytokine analysis

Across all measured molecules, only a minority followed the assumption of normality. Consequently, we relied on the Mann–Whitney-U test to detect significant differences in marginalized NPX values between our study cohorts. The test $p$ values were corrected using a BH procedure to control the FDR.

## CSF/plasma ratio analysis

We investigated the relative concentration of each molecule between CSF and plasma (CSF/plasma index) to investigate whether they result from intrathecal or peripheral synthesis. We first performed a median normalization of each molecule's NPX value. Significant differences in average ratios between groups were assessed using a Mann–Whitney-U test with a BH correction to control for the FDR.

## Association of the WHO progression scale with soluble proteins

We assessed the association between soluble proteins and COVID-19 severity based on the WHO progression scale[14] using a complement of four models. The first two models corresponded to ordinal regression models with an L1 norm, respectively with a backward and forward formulation, as implemented in the glmnetcr R package [https://cran.r-project.org/web/packages/glmnetcr/citation.html]. The lambda hyper-parameter (corresponding to the strength of the regularization term), was selected to minimize the Bayesian Information Criterion, or the Akaike Information Criterion when the BIC pointed to the null model. While this modeling approach is appropriate to the nature of the data, the absence of several categories from our sample as well as the poor predictive performance of the predictive model led us to complement this approach with simple linear regression models. We fitted Gaussian linear models with an L1 norm, using a 10-fold cross-validation, as implemented in the glmnet R package [https://cran.r-project.org/web/packages/glmnet/citation.html]. We kept the model, which minimized the cross-validated mean squared error and the

most-regularized model such that the cross-validated error is within one standard error of the minimum. We assessed the assumptions of these Gaussian models and found them appropriate. Finally, we cross-referenced the results from each of the four models (backward-ordinal, forward-ordinal, best linear, most-regularized ordinal) to provide a robust modeling choice of sets of molecular markers of COVID-19 severity.

## CSF-plasma and brain volumetric correlation analysis

The correlations between plasma and CSF measurements (between fluid correlations) were measured, for each COVID-19 class and control group, using the Kendall rank correlation coefficient (Kendall's tau).

To correlate between CSF or plasma molecules and brain region volumes, we selected the brain regions and molecules, which differed significantly between COVID-19 and control patients. A Spearman-rank correlation test was performed to assess the association between regional brain volumes and marginalized protein expression levels in CSF and plasma. The BH procedure was used for FDR control.

## ROC-AUC analysis and random forest approach

We ranked molecules by the ROC-AUC score of their marginalized data to discriminate against either: class I vs. III, class II vs. III, class I + II vs. III (Supplementary Data 3). Across all molecules, only a minority was normally distributed. Consequently, we relied on the nonparametric Mann–Whitney-U test to detect significant differences between each group. Individual $p$ values were corrected using the BH procedure, controlling for an FDR of 0.05. Complementary to this univariate approach, we assessed the relative importance of each molecule in a multivariate approach by training a random forest. Hyper-parameters were optimized using leave-one-out cross-validation. We performed this procedure twice: once with both CSF and plasma measurements, and once with only the plasma measurements. In both cases, the best random forest was able to predict the training set with perfect accuracy. We wish to remark explicitly that given our sample size, we kept the entirety of the data as a training set and report the testing set.

## Association of long-COVID with soluble proteins

We ranked molecules by the AUC-ROC score of their marginalized data to discriminate between patients who developed long-COVID and patients who did not. Subsequently, we trained logistic regression models with, respectively, the plasma and CSF molecules whose individual AUC-ROC was above 0.75. The logistic regression models were reduced using backward selection in order to limit overfitting and provide minimal sets of molecules sufficient to predict long-COVID. AUC-ROC cut-offs applied for minimal sets of predictive molecules were 0.7. The reduced model contained 4 plasma and 3 CSF molecules. Given our sample size, we kept the entirety of the data as a training set and reported the testing set. However, we assessed the robustness of our reduced sets of molecular markers using 5-Fold cross-validation.

**Brain imaging.** All uses of prospectively obtained brain scans have been approved by the Ethics Committee of Northwestern and Central Switzerland (clinicaltrials.gov NCT04472013, IRB approval EKNZ 2020-01503). Written informed consent has been provided by each participant. If participants were not able to provide consent their relatives provided informed written consent. Imaging studies were conducted on a 1.5 Tesla (T) MAGNETOM Siemens Avanto Fit and a 3 T MAGNETOM Siemens Skyra Scanner. MRI sequences included 3D T1-weighted (T1w) +/− gadolinium, fluid-attenuated inversion recovery (FLAIR), diffusion-weighted imaging (DWI), susceptibility-weighted imaging (SWI) and T2-weighted (T2w) sequences to document signs of neuroinflammation. For standardization of MRI interpretation, an assessment protocol was created (Supplementary Data 9). Anatomical T1w

MPRAGE pulse sequences were acquired for brain volumetric analysis. CT scans were assessed according to clinical standards. Two neuroradiologists (J.M.L. and M·N.P.) reviewed the images, blinded to clinical and laboratory patient data.

## Brain volumetric analysis

**Participants and imaging data acquisition.** Among the 40 enrolled COVID-19 patients (mean [SD] age, 54 [20] years; 17 women (42%), 22 were selected based on the 3D high-resolution T1w anatomical image quality. To generate a bigger sample size, 13 additional patients who underwent brain MRI during their acute phase of COVID-19 were retrospectively added. These patients were not included in the main study cohort, undergoing CSF and plasma analysis. Consequently, the COVID-19 volumetric imaging cohort consisted of 35 participants (mean [SD] age, 52 [20] years; 21 women (60%).

As a control cohort, 36 healthy, age- and sex-matched individuals were selected (mean [SD] age, 54 [24] years; 23 women (64%). This control group served only for the imaging analyses and was a different group than the healthy controls for the CSF and plasma comparison. The control groups' demographic and clinical information can be found in Supplementary Table 5. The acquisition of prospectively collected brain scans was approved by the Ethics Committee of Northwestern and Central Switzerland (clinicaltrials.gov NCT04472013, IRB approval EKNZ 2020-01503). For retrospectively obtained brain scans, the general research consent under local hospital regulations was acquired.

The 3D high-resolution T1w anatomical images were acquired using two MRI scanners (scanner 1: 1.5 T Siemens Avanto Fit; scanner 2: 3 T Siemens Skyra). An MPRAGE pulse sequence covering the whole brain was used in both MRI scanners with the following parameters. Scanner 1: 160 contiguous slices of 1 mm thickness in sagittal orientation; in-plane FOV = 256 × 256 mm$^2$, and matrix size 256 × 256 yielding an in-plane spatial resolution of 1 × 1 mm$^2$ and voxel size of 1 × 1 × 1 mm$^3$. The echo (TE), repetition (TR), and inversion (TI) times were set to TE/TR/TI = 2.8 ms/2400 ms/900 ms with a flip angle FA = 8°. Scanner 2: 160 contiguous slices of 1 mm thickness in sagittal orientation; in-plane FOV = 256 × 240 mm$^2$, and matrix size 256 × 240 yielding an in-plane spatial resolution of 1 × 1 mm$^2$ and voxel size of 1 × 1 × 1 mm$^3$. The echo, repetition, and inversion times were set to TE/TR/TI = 2.98 ms/2300 ms/900 ms with a flip angle FA = 9°. For the association analysis with the brain's regional volume, we included routine diagnostic CSF parameters (leukocytes, lactate, protein, and CSF/blood albumin ratio). These variables were available for the main COVID-19 study cohort, which was under CSF and plasma evaluation.

## Data pre-processing: brain global, regional gray matter and choroid plexus volume computation

The anatomical T1w images were automatically parcellated into 132 brain regions based on Neuromorphometrics atlas using the Neuromorphometrics toolbox. The atlasing methodology consists of two main steps. First, each image is segmented into three different brain tissue classes (CSF, gray matter, and white matter) using the "Segment" (unified segmentation) tool in SPM12 (Statistical Parametric Mapping Toolbox), which includes registration to the MNI (Montreal Neurological Institute) space. Second, the probabilistic atlas of each of the anatomical structures is spatially registered with the extracted gray and white matter tissue maps using the "Shoot" tool in SPM12, based on a nonlinear advanced registration algorithm[78]. Rules of probability are used to combine the previous images to obtain a probabilistic label map for each brain structure. At every gray matter voxel (in subject space), the probability of belonging to a specific anatomical structure is provided. From above, maximum probability label maps are calculated at all gray matter voxels (in subject space) which are labeled according to the structure of maximum probability. Finally, mean gray

matter volume (GMV) values are calculated across voxels belonging to each structure label (Supplementary Data 10). The total intracranial volume (TIV) was computed as the sum of gray and white matter and cerebrospinal fluid volumes in $cm^3$. Normalized GMV is defined as the ratio between GMV and TIV. To control for head size, we adjusted the statistical models for TIV measured by SPM12.

## Statistical analyses

We checked the normal distributions of all variables using Shapiro-Wilk tests and visual inspection of the histograms. To test the equality of variances, Levene's test was applied. Clinical and demographic variables were compared between groups with Independent t-test, Mann–Whitney-U test, or Chi-square tests where appropriate. Regional volumes were compared between groups using a linear regression model. The additional covariates were age, sex, age*sex interaction, MRI magnetic field strength, and TIV. Choroid plexus volume (CPV) was adjusted for TIV and was compared by Mann–Whitney-U test. We checked whether the dependent variable's variance is equal between the groups by performing Levene's test of equal variances. The $p$ values were adjusted for multiple comparisons using FDR. The associations between brain regional volume and clinical measures were assessed using partial correlation. The method allows calculating the linear partial correlation between our variables of interest adjusting for different covariates. Our covariates were: age, sex, age*sex interaction, MRI magnetic field strength, and TIV. We adjusted for multiple comparisons using an FDR method. The statistical analysis was performed using the JASP software (https://jasp-stats.org/). For the partial- correlation analysis, we used MATLAB software ('partialcorri.m' function) (https://www.mathworks.com/).

## Reporting summary

Further information on research design is available in the Nature Portfolio Reporting Summary linked to this article.

## Data availability

All data underlying the study are available within the submitted paper. No restrictions on data availability such as a materials transfer agreement are foreseen for this study. The trial protocol is available upon request from the corresponding author. The final trial protocol is provided as a Supplementary File (Supplementary Data 11). Source data are provided with this paper as an.xlsx file with specified different sheets corresponding to the Main Figures, Supplementary Figures and Supplementary Tables provided with this paper. The following external databases and datasets were used: SPM software as a suite of MATLAB (MathWorks) functions and subroutines with some externally compiled C routines (Wellcome Trust Center for Neuroimaging, London, UK; http://www.fil.ion. ucl.ac.uk/spm), Neuromorphometric atlas (SPM12 introduces a new atlas "labels_Neuromorphometrics"; see https://github.com/neurodebian/spm12/blob/master/spm_templates. man, http://Neuromorphometrics.com/). Maximum probability tissue labels derived from the "MICCAI 2012 Grand Challenge and Workshop on Multi-Atlas Labeling" are available in files tpm/labels Neuromorphometrics.{nii,xml}. This data was released under the Creative Commons Attribution-NonCommercial (CC BY-NC) with no end date. The MRI scans originate from the OASIS project and the labeled data as "provided by Neuromorphometrics, Inc. under academic subscription". Source data are provided with this paper.

## Code availability

All statistical reports are available under https://github.com/WandrilleD/severe-neuro-COVID-cross-sectional-study-etteretal2022. The molecular data analysis was realized using python tools and libraries (Python Software Foundation, http://www.python.org). Statistical computations relied on the numpy, pandas, scikit-learn, scipy and statsmodels libraries. Figures were generated using the graphviz, matplotlib and seaborn libraries.

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

## Acknowledgements

We thank the patients and their relatives for participating in this trial. We thank the medical and nursing staff of the University Hospital Basel Intensive Care Unit and COVID-19 cohort wards, as well as the Neuroradiology and Neurosurgery Departments for their efforts during this trial. We thank Klaus Dornmair for providing us with the h818C5 monoclonal antibody, Christopher Linington for providing us with the A12/18.1 monoclonal antibody, Hedda Wardemann for providing us with the polyreactive ED-38 antibody clone and Emma Allen-Vercoe for providing us with the microbial template. This study was funded by the BOTNAR Fast Track Call foundation grant (FTC-2020-10) awarded to G.H., M.M. and A.T.; the Swiss National Science Foundation Professorial Fellowship (PP00P3_176974), the ProPatient Forschungsstiftung, University Hospital Basel (Annemarie Karrasch Award 2019), and the Department of Surgery, University Hospital Basel, to G.H. The Neuroimmunology and Multiple Sclerosis Research Section, Department of Neurology, University Hospital Zurich, Zurich, Switzerland, with support by the Clinical Research Priority Program (CRPP) MS as well as the CRPP Precision-MS of the University Zurich, Zurich, Switzerland. The work was partly supported by the Swiss National Science Foundation (grant number: 4078P0_198345, title: Protective and pathogenic T-cell immunity during SARS-CoV-2 infection) and the Loop Zurich, COVID-19 project (SARS-CoV-2-induced immune alterations and their role in post-COVID syndrome). The study was also funded by the Propatient Foundation (pp200526), the Swiss National Science Foundation (Eccellenza Fellowship; PCEFP3_194609), National Multiple Sclerosis Society (Kathleen C. Moore Fellowship; FG1912-35229 and the Goldschmidt-Jacobson Foundation grants awarded to A-K.P. The sponsor (BOTNAR fast Track Call foundation; grant: FTC-2020-10) had no role in the design of the study, data collection and analysis, data interpretation or writing of the paper.

## Author contributions

Conceptualization: G.H., MM.E. Methodology: G.H., MM.E., TA.M., L.K., E.P., A-K.P. Clinical data and patient integration: G.H., MM.E., TA.M., J.R., N.E., A.C., C.E., I.J., E.K., H.P., M.S., A-K.P., GS. D., LM.G., J.M., Ö.Y., U.S., L.K. Sample handling and processing: MM.E., TA.M. Software: W.D., S.H. Formal analysis: W.D., S.H., JM.L., M-N.P., G.H., MM.E, TA.M., E.P., L.K., A-K.P. Investigation: MM.E., TA.M., L.K., E.P. Resources: N.E., J.R., M.S., M.K., I.J., H.P., M.S., J.O., J.K., R.G., M.L., HH. H. Writing—original draft: G.H., MM.E., TA.M., L.K., E.P., A-K.P. Writing—review and editing: all authors. Supervision: G.H., C.G., A-K.P., M-N.P. Project administration: G.H., A.T., M.M. Funding acquisition: G.H., A.T., M.M.

## Competing interests

I.J. has received speaker honoraria or unrestricted grants from Biogen Idec and Novartis and has served as an advisor for Alexion, Biogen, Bristol Myers Squibb, Celgene, Janssen-Cilag, Neuway, Merck, Novartis, Roche and Sanofi Genzyme; none of these are related to this study. G.H. has equity in and is a co-founder of Incephalo Inc. A-K.P. has received speaker honoraria or research/travel support from Roche and Biogen all used for research. The remaining authors declare no competing interests.

## Additional information

[1]Brain Tumor Immunotherapy Lab, Department of Biomedicine, University of Basel, Basel, Switzerland. [2]Departments of Neurology, Biomedicine and Clinical Research, & Research Center for Neuroimmunology and Neuroscience Basel, University Hospital Basel, University of Basel, Basel, Switzerland. [3]sciCORE Center for Scientific Computing, University of Basel, Basel, Switzerland. [4]Translational Imaging in Neurology (ThINK) Basel, Department of Medicine and Biomedical Engineering, University Hospital Basel, University of Basel, Basel, Switzerland. [5]Department of Neurology, University Hospital Basel, Basel, Switzerland. [6]Department of Neurosurgery, University Hospital Basel, Basel, Switzerland. [7]Department of Surgery, University Hospital Basel, Spital-strasse 21, Basel, Switzerland. [8]Neurosurgical Intensive Care Unit, Department of Neurosurgery and Institute of Intensive Care Medicine, University Hospital Zurich, Zurich, Switzerland. [9]Neuroimmunology and Multiple Sclerosis Research Section, Department of Neurology, University Hospital Zurich, Zurich, Switzerland. [10]Department of Intensive Care Medicine, University Hospital Basel, Basel, Switzerland. [11]Department of Pathology, Institute of Medical Genetics & Pathology, University Hospital Basel, University of Basel, Basel, Switzerland. [12]Department of Neuroradiology, Clinic for Radiology & Nuclear Medicine, University Hospital Basel, Basel, Switzerland. [13]Division of Medical Immunology, Laboratory Medicine, University Hospital Basel, Basel, Switzerland. [14]Clinical Virology, University Hospital Basel, Basel, Switzerland. [15]Transplantation & Clinical Virology, Department of Biomedicine, University of Basel, Basel, Switzerland. [16]Infectious Diseases & Hospital Epidemiology, University Hospital Basel, Basel, Switzerland. ✉e-mail: gregor.hutter@usb.ch

