## [Peer Review File · Nature Communications]

Severe Neuro-COVID is associated with peripheral immune signatures, autoimmunity and neurodegeneration: a prospective cross-sectional studyREVIEWER COMMENTS

Reviewer #1 (Remarks to the Author):

I had the pleasure to review the manuscript entitled "Severe Neuro-COVID is associated with peripheral immune signatures, autoimmunity and neurodegeneration: a prospective cross-sectional study". The manuscript is well written and appropriately organized. I would like to invite the authors to revise their manuscript considering the following points:

Major point:

- Unfortunately the imaged patients do not have an internal control (pre-COVID MR). In fact, it is not clear if the MRI findings are essentially related to COVID-19 infection or merely due to other causes, such normal aging. It is well known that older patients have T2 hypersignal intensities involving periventricular and subcortical white matter, which can be due to chronic ischemic or age related degenerative changes. In your study population, there are several cases of above 70 years old (up to 98 years old). Unless you have a baseline (pre-COVID) brain MR and compare it with post-COVID imaging, it is very difficult to convince people all these changes are relevant to COVID-, not the aging process.

- Again, I am not sure how we can make sure the brain CT findings of stroke or intracranial hemorrhage are related to COVID-19, not the aging process. CT is limited in aging of stroke.

Minor points:

- Although you briefly mention the need for further studies, the statement is rather general. I suggest elaborating on this statement with a few directions for future research that are supported by your current results and would benefit the field.

Reviewer #2 (Remarks to the Author):

Neurological manifestations of COVID-19 have been termed Neuro-COVID and are considerably more prevalent than neurological sequelae in infections due to related coronaviruses. Better understanding its mechanisms is thus relevant for the the pandemic but also more generally for neurological sequelae of viral infections.

Here, Etter et. al. investigated Neuro-COVID using CSF and serum proteomics and imaging data across different Neuro-COVID severity classes. In comparison to other CSF studies of Neuro-COVID, the patient numbers and the number of molecules studied are high. The combination of imaging and proteomics techniques with clinical characterization is particularly interesting. Although this is a highly relevant study that adds substantially to our understanding and recruited an impressive number of patients, the manuscript falls short of its potential. I often missed a clear thread throughout the manuscript, although the visualization in Figure 5 is somewhat helpful. Also, more could be gained from the available data by extending the statistical analysis. There are several issues that need to be addressed by the authors in a major revision:

Main points:

Some data indicate that Post-/Long-COVID (also named PASC) is the new epidemic after the COVID-19 pandemic. Post-/Long-COVID is diagnosed if sequelae persist after one or two months depending on the CRC/WHO definition. In the present study, most of the non-deceased Neuro-COVID patients were re-evaluated after 10 months. And the authors could therefore test for predictive potential of the biomarkers collected during the acute phase for predicting the severity or frequency of PASC. This could considerably improve the impact of the manuscript

I am disappointed by what the authors name a correlation analysis. On page 8, lines 147-149 the authors conclude that there might be a correlation between Neuro-COVID severity and CSF proteomics. But why did they not quantify this correlation for example by numerically correlating clinical COVID-19 severity scores (e.g. Severity Index) or laboratory surrogates of COVID-19 severity (e.g. IL-6) with the concentration CSF proteins? Then this statement would be backed by quantitative data.

Related to the first two points, on page 10, 1st paragraph the authors aim for CSF / plasma correlation. But from my perspective this mainly helps our understanding of how the CSF compartment is separated (or connected) to the plasma. In line 197 they introduce the concept by stating 'To forecast severe Neuro-COVID....' But why does a CSF/plasma correlation help a clinical forecast? This would need a correlation between proteomics results with clinical severity scores at 10 months (e.g. mRS, Karnovsky index, etc).

I appreciate the PCA approach, however Figure 1B shows a lot of overlap and the results of this plot are not discussed in the results. Maybe other dimensionality reduction techniques (eg t-SNE, UMAP) show a clearer separation?

I think the manuscripts could be significantly improved by adding data to corroborate the finding of a polyclonal B cell response. Optimally, this would be bulk BCR/TCR of CSF and blood of all samples. Are there differences across severity and are clones shared between Neuro-COVID patients and compartments?

The results section of the manuscript would benefit from short single sentences at the beginning of each paragraph introducing why the respective analysis was performed. Similarly, one could conclude from the data that many of the abnormalities observed in CSF in severe Neuro-COVID are peripherally initiated. This supports the peripheral immune hypothesis of Neuro-COVID (i.e. peripheral cytokine storm unspecifically affects brain function). But specific mechanisms (e.g. anti-neuronal autoimmunity, T cell exhaustion occurring in the CSF only in Neuro-COVID) have been proposed. I cannot find appreciation of this context.

It is very hard to find the information about what exactly the inflammatory control patients suffered from. Although it can be deduced from Suppl. Tab. 9, it would help the reader to know earlier that these are mostly infectious, not autoimmune conditions. Accordingly, I do not consider the label inflammatory controls as adequate. Maybe 'CNS inflammatory' or infectious would be more appropriate.

The study protocol in file 355823_0_supp_6316501_r7pgm1 mentions CyTOF of blood cells. Is this being published elsewhere or are the data unavailable?

The reader would benefit from a schematic plot localizing the differentially abundant proteins to immune cell types mainly producing them. For example, CLEC9A could also be considered a marker of cDC1.

Both MRI paragraphs on pages 11 and 12 conclude by saying that no comparisons were significant. I lack the background to judge the imaging aspects behind this, but are any of the changes real or are they not? This is somehow left for the reader to decide. Also, the findings are poorly summarized. What is the key message of these paragraphs? How do the findings compare to previous reports of cortical hypometabolism in PASC by PET imaging?

In Figure 3d the authors tested individual parameters for diagnostic power. Can the individual parameters be prioritized? For example using feature selection approaches (e.g. random forest, Lasso etc)? Which parameters are most powerful? What would be a minimal set of features?

Minor points:

There is 1 multiple sclerosis patient in the class I COVID-19 patients. Since those patients are compared to inflammatory controls, this could bias the results. This should be discussed.

In Figure 1A and C, some comparisons are marked as significant by asterixis, some as non significant, but several comparisons (e.g. healthy controls vs Class III) seem to miss. Were those comparisons not tested (for what reason?) or does this represent non significance? I think it would be easier to only show the significant comparisons and mention in the legends that all other comparisons were non significant.

The authors claim that CSF glucose is elevated in Neuro-COVID patients. This should be replaced

with the glucose ratio, especially since class III has a relevant proportion of Type 2 diabetes (53,3%), and a blood-brain barrier disruption. How would the author interpret an elevated CSF/serum glucose ratio in the context of Neuro-COVID?

Panels in Figure 2a are not aligned, labels are too small to be readable, all boxplots could be considerably more narrow without losing information and groups should not only be labeled by color but also by x-axis labels.

The authors state that "Principal component analysis (PCA) of soluble proteins distinctly separated controls from COVID-19 patients". If you look at the ellipses, which, as the authors state, represent the 95% confidence interval, this is only true for class II in plasma.

The paragraph "Neuro-COVID class III features are manifestation of microglia regulation, neurodegeneration and blood-brain barrier disruption" relies on Supplementary Fig. 4. I think the main findings should be shown in the main figures.

Figure 3C shows in the dendrogram that class II is distinct and that class I and class III are even closer to healthy controls and inflammatory controls than to class II. This seems quite surprising, any explanations? How would the authors contextualize this distinct profile (e.g. featuring T cell cytotoxicity molecules CD8A, GZMA)?

It's misleading if the authors state that "gray matter volumes in olfactory pathway structures decrease in Neuro-COVID patients" as a main finding in a subheader, but then write that the differences did not pass the significant threshold when adjusted for multiple testing

Fig. 4h-j are not mentioned in the results.

The statement on page 4, lines 82/83 'However, ... has rarely been detected in the CSF...' needs references. There are several.

Reviewer #3 (Remarks to the Author):

This is an interesting study by Dr Etta et al aiming to better understand disease mechanisms in neuro-covid. It has several strengths, in particular the comprehensive array of biomarkers they examine in blood and spinal fluid, and the link to imaging modalities; the data on auto-antibodies are interesting. Much of what they found confirms previous findings in terms of biomarkers elevated in serum and spinal fluid, what kind of changes are seen on brain imaging, et cetera. The results indicating the TRANCE/RANKL production in the CNS were interesting. The potential link between specific biomarkers and brain regions is interesting.

1. the main limitation of the paper is that the number of subjects is really quite small, and they did not in the analyses account for confounding factors such as age and comorbidity. The total number studied was 40, of whom just 15 were classified as having the most severe disease. There were some significant differences found, but then this is not surprising given the number of parameters looked at and the number of comparisons made. Perhaps having an initial hypothesis might have reduced the numbers of parameters assessed.
2. They did well to compare with healthy controls and with inflammatory controls; but it was surprising they did not compare with covid patients who did NOT have CNS disease; without doing this, how do we know to what extent the findings are due to neurological covid disease, as opposed to covid itself?
3. In terms of numbers of patients, it was surprising so many patients were approached and did not agree to participate. It was unclear on what basis patients were recruited - ie what were the entry criteria to the study. I looked in the Trial protocol which was helpfully included, but I was not really much the wiser (incidentally the protocol said just 20 patients and 20 controls would be recruited, but the numbers went beyond this; I presume there were later amendments which were approved).
4. The paper is quite confusing to follow.

There are subheadings, such as:

"Gray matter volumes in olfactory pathway structures decrease in Neuro-COVID patients and negatively correlate with inflammatory CSF parameters"

But the text then says:

However, this finding was not significant after FDR correction.

And:

"High PD-240 L1 and HGF plasma levels are associated with decreased regional gray matter volumes, while GDF-8 and BMP-4 are neuroprotective in Neuro-COVID patients"

But then the text says:

"none of the p-values were significant after BH-procedure."

If the purported values are not significant after correction, then why the declarative sub-heading?

5. The team classified their patients into three groups referencing the paper by Fotuhi et al. However, the Fotuhi paper really just comes up with a biological hypothesis about what might be happening in patients, and grades them into three levels, but there is not a great deal of evidence that this hypothesis is correct. The Fotuhi hypothesis assumes a single disease mechanism which explains a continuum of disease severity from I to III, resulting in the different presentations. Although we would all like to believe this, because it makes understanding the disease easier, it may well not be the case! It may be that the disease mechanism causing mild encephalopathy in some patients is completely different to the disease mechanism causing strokes in other patients.
6. The authors describe the patients as "clinically well-characterised" but unless I missed it, there was very little clinical characterisation. What were the clinical features that led to the patient classification into grades I, II, III? One of the patients with the mildest disease (grade I) died. What happened here? How do they make sense of this?
7. The Fotuhi paper does not actually describe how patients should be defined clinically into their three groups, so how did the current authors do this? We get almost no clinical information, such as how many were encephalopathic, how many had strokes, what type, etc etc

In summary, then Etti et al make fabulous use of the samples and imaging they collected, but the numbers are too small to be sure that their findings are meaningful, and the presumption of a severity continuum, as per the hypothesis in the Fotuhi paper limits the usefulness of the interpretation.

Reviewer #1 (Remarks to the Author):

I had the pleasure to review the manuscript entitled "Severe Neuro-COVID is associated with peripheral immune signatures, autoimmunity and neurodegeneration: a prospective cross-sectional study". The manuscript is well written and appropriately organized. I would like to invite the authors to revise their manuscript considering the following points:

Major point:
- Unfortunately the imaged patients do not have an internal control (pre-COVID MR). In fact, it is not clear if the MRI findings are essentially related to COVID-19 infection or merely due to other causes, such normal aging. It is well known that older patients have T2 hypersignal intensities involving periventricular and subcortical white matter, which can be due to chronic ischemic or age related degenerative changes. In your study population, there are several cases of above 70 years old (up to 98 years old). Unless you have a baseline (pre-COVID) brain MR and compare it with post-COVID imaging, it is very difficult to convince people all these changes are relevant to COVID-, not the aging process.

We thank reviewer 1 for his concerns regarding pre-COVID MRIs. We totally agree and are aware of our several patients aged older than 70 and subsequent consequences on structural brain imaging. Patient recruitment for this trial was difficult, mainly because of our need for a lumbar puncture, which led a majority of screened patients to refuse their participation. Adding a pre-existing MRI exam as an additional requirement for study participation would have made patient accrual in this prospective setting almost impossible. We therefore describe the structural MRI changes without implying a causal relationship to COVID-19, and have highlighted this limitation in the discussion on lines 451-453. However, we indeed identified 6/40 patients with pre-existing, sequence comparable brain MRIs in our cohort (3 patients in class I and 3 patients in class III). Also, one class III patient underwent MRI during COVID-19 and one week later a cranial CT, which we also compared as far as possible using two different imaging modalities. We have compared the pre-COVID-19 MRI scans (diffusion weighted imaging and FLAIR/T2-weighted imaging) with their respective acute COVID-19 MRIs. The findings are described in the section "*Neuro-COVID class III patients feature striking findings on brain imaging while most class I and II patients lack evidence of neuroinflammation*" on lines 314-321. In 3 patients with pre-COVID-19 brain scans no changes occurred. In one patient (class I) we were able to detect signal alterations on diffusion weighted imaging, speaking in favor of acute cerebral diffusion restriction. In 2 other patients (one class I and one class III patient) we observed FLAIR/T2-weighted hyperintensities, which were not observable in the pre-COVID-19 MRI. The one class III patient undergoing MRI and cranial CT scans during acute COVID-19 (first MRI scan and one week later cranial CT) suffered acute cerebellar infarction and frontal subarachnoid hemorrhage. However, we are aware of the fact that we cannot attribute a causal relationship to COVID-19 in this small sample size and particularly in the small population with pre-COVID-19 brain scans. Therefore, we stated our findings in a descriptive manner without drawing any conclusions. In view of recent literature, however, these findings suggest a possible association with COVID-19.

- Again, I am not sure how we can make sure the brain CT findings of stroke or intracranial hemorrhage are related to COVID-19, not the aging process. CT is limited in aging of stroke.

We agree and we are aware of not drawing any conclusions and causalities. These findings are only a description of what we observed. Therefore, we discussed this point detailed in the discussion under "*Structural brain imaging alterations dominate in class III patients*" on line 454-461 and made sure that we did not implicate to make any causal relations between COVID-19 and these pathologies.

Minor points:

- Although you briefly mention the need for further studies, the statement is rather general. I suggest elaborating on this statement with a few directions for future research that are supported by your current results and would benefit the field.

We agree and therefore made the suggested statement in the discussion under “Structural brain imaging alterations dominate in class III patients” on line 457-462. We highlight the need for longitudinal imaging studies of COVID-19 affected individuals, in conjunction with targeted biomarker evaluation, to find means of interfering with the long-term effects of the disease.

Reviewer #2 (Remarks to the Author):

Neurological manifestations of COVID-19 have been termed Neuro-COVID and are considerably more prevalent than neurological sequelae in infections due to related coronaviruses. Better understanding its mechanisms is thus relevant for the the pandemic but also more generally for neurological sequelae of viral infections. Here, Etter et. al. investigated Neuro-COVID using CSF and serum proteomics and imaging data across different Neuro-COVID severity classes. In comparison to other CSF studies of Neuro-COVID, the patient numbers and the number of molecules studied are high. The combination of imaging and proteomics techniques with clinical characterization is particularly interesting. Although this is a highly relevant study that adds substantially to our understanding and recruited an impressive number of patients, the manuscript falls short of its potential. I often missed a clear thread throughout the manuscript, although the visualization in Figure 5 is somewhat helpful. Also, more could be gained from the available data by extending the statistical analysis. There are several issues that need to be addressed by the authors in a major revision:

We thank reviewer 2 for his critical and constructive comments on our study. We hope to address your concerns in the revised version of our manuscript encompassing a substantially improved and enhanced analysis of the trial data.

Main points:

Some data indicate that Post-/Long-COVID (also named PASC) is the new epidemic after the COVID-19 pandemic. Post-/Long-COVID is diagnosed if sequelae persist after one or two months depending on the CRC/WHO definition. In the present study, most of the non-deceased Neuro-COVID patients were re-evaluated after 10 months. And the authors could therefore test for predictive potential of the biomarkers collected during the acute phase for predicting the severity or frequency of PASC. This could considerably improve the impact of the manuscript

This is a very good point and could indeed significantly improve the power of the manuscript. We consequently tested for predictive biomarkers to potentially forecast long-COVID. We defined long-COVID according to the World Health Organization definition (WHO/2019-nCoV/Post_COVID-19_condition/Clinical_case_definition/2021.1) and created two groups (long-COVID versus non-long-COVID). Also, we performed a further follow-up (now 13-months follow-up) to account for any changes during the last 3 months that passed since the 10-months follow-up. The results are described in the results subsection “Long-COVID is more prevalent in severe Neuro-COVID patients and associated with specific CSF and plasma parameters” on the following lines 346-365, and discussed in the subsection “Long-COVID is predicted by a peripheral and CNS innate immune dysregulation” on lines 498-509, and visualized in the newly created Figure 8. Briefly, we identified a signature of 3 CSF and 4 plasma biomarkers tightly

associated with long-COVID, using a multimodal modelling approach including AUC and logistic regression analysis as outlined in the method (lines 694-701) and code section (provided in the supplementary material) of the revised manuscript.

I am disappointed by what the authors name a correlation analysis. On page 8, lines 147-149 the authors conclude that there might be a correlation between Neuro-COVID severity and CSF proteomics. But why did they not quantify this correlation for example by numerically correlating clinical COVID-19 severity scores (e.g. Severity Index) or laboratory surrogates of COVID-19 severity (e.g. IL-6) with the concentration CSF proteins? Then this statement would be backed by quantitative data.

Thank you for bringing up this interesting point. Yet, we have assessed the correlation of CSF and plasma proteins differing across groups and the COVID-19 severity, which was assessed using the clinical WHO progression scale ("A minimal common outcome measure set for COVID-19 clinical research," 2020). Accordingly, we used a complement of four different models (backward-ordinal, forward-ordinal, best linear and most-regularized ordinal) to provide a robust set of CSF and plasma markers associated with different COVID-19 severity degrees. The results are described in the results section "*Mediators involved in microglia regulation, tissue damage and blood-brain barrier disruption are associated with a high WHO clinical progression scale score*" on lines 241-255. The corresponding Venn diagram is included in the Figure 6a, where the results of each used model is represented.

Related to the first two points, on page 10, 1st paragraph the authors aim for CSF / plasma correlation. But from my perspective this mainly helps our understanding of how the CSF compartment is separated (or connected) to the plasma. In line 197 they introduce the concept by stating 'To forecast severe Neuro-COVID....' But why does a CSF/plasma correlation help a clinical forecast? This would need a correlation between proteomics results with clinical severity scores at 10 months (e.g. mRS, Karnovsky index, etc).

Thank you for this comment. Indeed, this was not well enough explained and therefore we rephrased it on lines 258-261. We aimed at identifying CSF and plasma biomarkers that are associated with severe Neuro-COVID (AUC-ROC analysis of class I and II vs class III). Additionally, we aimed at identifying markers depicting a strong CSF-plasma correlation, because plasma samples are daily obtained in the clinics and therefore obtaining CSF by lumbar puncture would not be necessary then.

I appreciate the PCA approach, however Figure 2B shows a lot of overlap and the results of this plot are not discussed in the results. Maybe other dimensionality reduction techniques (eg t-SNE, UMAP) show a clearer separation?

Thank you for pointing this out. Indeed, we did not properly describe the results of the PCA plot in Figure 2B. We now described the result of the new plot. As requested, we tried another dimensionality reduction technique with UMAP, but we failed to show a better segregation of the different groups with this modality (PtP-Reply Figure 1). Therefore, we applied another dimensionality reduction technique using non-metric multidimensional scaling (NMDS) as outlined in new Figure 2b, lines 157-160.

PtP-Reply Figure 1: UMAP representation of merged ant-BSA, anti-ds-DNA and anti-gut bacteria antibodies in CSF and plasma per patient group fails to show clear separation.

I think the manuscripts could be significantly improved by adding data to corroborate the finding of a polyclonal B cell response. Optimally, this would be bulk BCR/TCR of CSF and blood of all samples. Are there differences across severity and are clones shared between Neuro-COVID patients and compartments?

This is certainly a valid suggestion. Unfortunately, CSF samples were not prepared technically to accomplish either bulk or single cell RNAseq analysis, since this would have necessitated fresh processing of the cellular fraction directly after acquiring CSF by lumbar puncture. However, for frozen PBMCs, we were able to bulk-sequence a selection of class I and class III patients for BCR clonality, using an Oncomine BCR IGH SR RNA assay (Thermo Fisher) to sequence Ig heavy chain complementarity determining region 3 (CDR3). Our initial plan for the revision was to perform scRNA seq of the PBMC fraction, however, the viability of the cellular fraction was suboptimal for this endeavour.

The results of this analysis are highlighted on lines 171-186, and visualized in new Fig. 3a-d. Patient details are summarized in supplementary table 3. We discuss and speculate on the findings of this analysis in the discussion on lines 408-414. Briefly, we find a polyclonal plasma B cell response in both class I and class III patients. However, the amount of B cell clones in the examined class I patients was higher, as well as the frequency of specific clones. Severely affected patients were not able to produce specific clones, but had an overall polyclonal response. Together with an enhanced blood brain barrier disruption in class III, this might lead to ingress of these antibodies into the CSF compartment (where we clearly see an increased antibody concentration).

Although an additional analysis of the other samples would potentially provide us with more information on the clonality of the B cells in different Neuro-COVID subclasses, we feel that the general hypothesis of the peripheral induction of B cell polyclonality in COVID-19 and subsequent (peripheral cytokine induced) CNS ingress might already be supported by these data. Research funding for a larger scale B- and T cell single cell analysis in both CSF and plasma compartments, potentially in a multicentric consortium, would be needed to decipher the overall implications of the observed phenomena.

The results section of the manuscript would benefit from short single sentences at the beginning of each paragraph introducing why the respective analysis was performed.

Thank you for this comment. As requested, we added short introductions to each result paragraph to clarify why the respective analysis was performed. We hope this helps structuring the manuscript better.

Similarly, one could conclude from the data that many of the abnormalities observed in CSF in severe Neuro-COVID are peripherally initiated. This supports the peripheral immune hypothesis of Neuro-COVID (i.e. peripheral cytokine storm unspecifically affects brain function). But specific mechanisms (e.g. anti-neuronal autoimmunity, T cell exhaustion occurring in the CSF only in Neuro-COVID) have been proposed. I cannot find appreciation of this context

As pointed out by the reviewer, our data clearly support the peripheral immune hypothesis. We show in many instances that mediators of anti-neuronal autoimmunity, T cell exhaustion, microglia activation and blood-brain barrier disruption can lead to the observed neurological effects. We hope to have clarified these mechanisms more clearly in the updated discussion part of the manuscript, and the summarizing Figure at the end of the manuscript in the discussion section (Fig. 9).

It is very hard to find the information about what exactly the inflammatory control patients suffered from. Although it can be deduced from Suppl. Tab. 9, it would help the reader to know earlier that these are mostly infectious, not autoimmune conditions.

We agree with the reviewer and have therefore described the conditions included in the inflammatory control group at the very beginning of the manuscript in section “*Clinical characteristics of the study cohorts and study interventions*”, line 126-130. Also, the table including the control groups’ clinical details has now been changed to Supplementary Table 1.

Accordingly, I do not consider the label inflammatory controls as adequate. Maybe ‘CNS inflammatory’ or infectious would be more appropriate.

As requested, we changed the label “inflammatory controls” to “CNS inflammatory” throughout the manuscript.

The study protocol in file 355823_0_supp_6316501_r7pgm1 mentions CyTOF of blood cells. Is this being published elsewhere or are the data unavailable?

Indeed, we planned on performing CyTOF of PBMCs. However, technical difficulties and personnel shortage precluded us from performing this analysis as of now. We decided to report potential results from a validated CyTOF panel separately, or in a larger, and potentially multicentric cohort of COVID patients.

The reader would benefit from a schematic plot localizing the differentially abundant proteins to immune cell types mainly producing them. For example, CLEC9A could also be considered a marker of cDC1.

Thank you for this suggestion. We created a schematic plot that localizes the differentially abundant proteins to immune cell types that mainly produce these proteins in Supplementary Figure 6 based on protein atlas data (www.proteinatlas.org, Figure S6, referenced on lines 290-292). We limited the amount of proteins to the most significant findings.

Both MRI paragraphs on pages 11 and 12 conclude by saying that no comparisons were significant. I lack the background to judge the imaging aspects behind this, but are any of the changes real or are they not? This is somehow left for the reader to decide. Also, the findings are poorly summarized. What is the key message of these paragraphs? How do the findings compare to previous reports of cortical hypometabolism in PASC by PET imaging?

We thank you for this comment. We have now expanded the results part of the imaging findings and tried to better explain the changes, also by adding the findings of patients with existing pre-COVID-19 MRIs. Because the signal alterations on brain imaging in class I and II patients are not specific for a single disease, we described the findings with caution and did not want to implicate any causalities of COVID-19 or SARS-CoV-2. Therefore, we also discussed these unspecific FLAIR/T2w imaging alterations in detail in the discussion under “*Structural brain imaging alterations dominate in class III patients*” line 448-456 and tried to make clear that there exists a broad spectrum of differential diagnosis explaining these structural brain imaging changes (amongst others inflammatory conditions and age).

Regarding the cortical hypometabolism: this is indeed a good and important point which we have now discussed under “*Standard CSF parameters associate with reduced olfactory GMVs in COVID-19 patients*” line 469-477. Neurodegeneration, as a result of loss of brain volume, is associated with cortical hypometabolism (low glucose uptake and turnover) on FDG-PET imaging. These findings have also been described in COVID-19 patients (Douaud et al., 2022). Since we found decreased gray matter volumes in our COVID-19 patients, mainly in brain regions belonging to the olfactory system (which in some regions extensively overlaps with memory-related functions), these results may point to neurodegenerative processes in severe COVID-19, resulting in these decreased regional brain volumes and COVID-19 related long-term neurological sequelae. This finding of ours is in line with the previously described hypometabolism, mainly in parietal and fronto-temporal regions (including olfactory pathways and memory-related functions of the brain), in COVID-19 patients.

In Figure 3d the authors tested individual parameters for diagnostic power. Can the individual parameters be prioritized? For example using feature selection approaches (e.g. random forest, Lasso etc)? Which parameters are most powerful? What would be a minimal set of features?

The individual parameter's AUC-ROC metrics can, and is, used to prioritize each parameter. However, to complement this individual approach we followed the reviewer's constructive advice and evaluated the parameters' relative importance in a multivariate framework.

We used a random forest approach, using a leave-one-out cross-validation approach to optimize hyper-parameters. We then contrasted the relative parameters importance in the univariate (AUC-ROC) and multivariate (random forest) as highlighted in Figures 6c and d in order to pinpoint the most powerful ones (updated methods, lines 687-694). Regarding minimal sets of features, there are many sets of 3 or 4 features which, together, provide a "perfect" prediction of our data, however this is likely due to the relative low number of sample and high number of parameters leading to spurious correlations and over-fitting. Thus, we do not feel confident about the robustness of any of these minimal sets of predictors and prefer to provide the relative importances only.

Minor points:

There is 1 multiple sclerosis patient in the class I COVID-19 patients. Since those patients are compared to inflammatory controls, this could bias the results. This should be discussed.

Thank you for bringing up this point. In Table 1 we presented which patients underwent lumbar puncture. In this regard, we tried to highlight that the one patient suffering from multiple sclerosis did not undergo lumbar puncture, therefore not influencing our CSF findings. We additionally highlighted this in the text in the results section "*Clinical characteristics of the study cohorts and study interventions*", line 108-109.

In Figure 1A and C, some comparisons are marked as significant by asterixis, some as non significant, but several comparisons (e.g. healthy controls vs Class III) seem to miss. Were those comparisons not tested (for what reason?) or does this represent non significance? I think it would be easier to only show the significant comparisons and mention in the legends that all other comparisons were non significant.

We totally agree and made the requested changes. Significances were now tested between all study groups and only significant differences are highlighted in the figures. Non-significant comparisons are mentioned in the figure legends.

The authors claim that CSF glucose is elevated in Neuro-COVID patients. This should be replaced with the glucose ratio, especially since class III has a relevant proportion of Type 2 diabetes (53,3%), and a blood-brain barrier disruption. How would the author interpret an elevated CSF/serum glucose ratio in the context of Neuro-COVID?

Thank you for pointing this out. Indeed, this is a very important question which we have assessed now. The differing CSF/plasma glucose ratios are described under "*Class III patients have an impaired blood-brain-barrier and a polyclonal B cell response*" line 147-149, depicted in Table 1 and Figure 2a. Possibly underlying mechanisms are discussed in the discussion section under "*Standard CSF parameters associate with reduced olfactory GMVs in COVID-19 patients*" line 469-477.

Panels in Figure 2a are not aligned, labels are too small to be readable, all boxplots could be considerably more narrow without losing information and groups should not only be labeled by color but also by x-axis labels.

We aligned Figure 2a, narrowed all boxplots and tried to make the labels better readable. Additionally, the groups are now labeled by their names.

The authors state that "Principal component analysis (PCA) of soluble proteins distinctly

separated controls from COVID-19 patients". If you look at the ellipses, which, as the authors state, represent the 95% confidence interval, this is only true for class II in plasma.

We thank you for pointing this out. We rephrased this part and tried to clearly describe the protein pattern depicted in the PCA plots under "*Targeted proteomic analysis of CSF and plasma reveals a robust peripheral immune response in Neuro-COVID and a class III-specific signature*" line 202-209.

The paragraph "Neuro-COVID class III features are manifestation of microglia regulation, neurodegeneration and blood-brain barrier disruption" relies on Supplementary Fig. 4. I think the main findings should be shown in the main figures.

We have transitioned the results from "Supplementary Figure 4" to the main figure panel (Figure 5).

Figure 3C shows in the dendrogram that class II is distinct and that class I and class III are even closer to healthy controls and inflammatory controls than to class II. This seems quite surprising, any explanations? How would the authors contextualize this distinct profile (e.g. featuring T cell cytotoxicity molecules CD8A, GZMA)?

Thank you for bringing up this point which is really interesting. We interpreted this finding as a possible sign of T cell exhaustion during disease progression, which could be the result of repetitive overstimulation due to immune dysregulation in class III patients. In class II patients, the immune dysregulation and inflammatory cascades are not as strong as in the most severe Neuro-COVID class, possibly resulting in a preserved and therefore still adequate T cell function. We integrated this possible explanation in the corresponding result section "*CSF-plasma correlations identify a neuronal damage signature in class III, encompassing predictive markers for severe Neuro-COVID*", lines 276-278.

It's misleading if the authors state that "gray matter volumes in olfactory pathway structures decrease in Neuro-COVID patients" as a main finding in a subheader, but then write that the differences did not pass the significant threshold when adjusted for multiple testing.

Thank you for pointing this out. This is an important point from your side, since we did not state it clear enough. Gray matter volumes in olfactory and gustatory pathway structures were significantly ($p < 0.05$) negatively associated with the CSF leukocyte count, CSF protein levels and CSF/plasma albumin ratio, also after FDR correction. We therefore rephrased this section and made it clear which parameters did not pass the significance threshold and which parameters were significant on lines 325-343.

Fig. 4h-j are not mentioned in the results.

Thank you for this remark. We made sure that the findings integrated in the corresponding figures (Fig. 7h-j) are now mentioned and well described in the result section "*Lower GMVs in olfactory pathway structures in Neuro-COVID patients are negatively correlated to inflammatory CSF parameters*", lines 322-343.

The statement on page 4, lines 82/83 'However, ... has rarely been detected in the CSF...' needs references. There are several.

We thank you for this comment. We have now added three references: (Bellon et al., 2021; Heming et al., 2021; Neumann et al., 2020)

Reviewer #3 (Remarks to the Author):

This is an interesting study by Dr Etta et al aiming to better understand disease mechanisms in neuro- covid. It has several strengths, in particular the comprehensive array of biomarkers they examine in blood and spinal fluid, and the link to imaging modalities; the data on auto-antibodies are interesting. Much of what they found confirms previous findings in terms of biomarkers elevated in serum and spinal fluid, what kind of changes are seen on brain imaging, et cetera. The results indicating the TRANCE/RANKL production in the CNS were interesting. The potential link between specific biomarkers and brain regions is interesting.

1. the main limitation of the paper is that the number of subjects is really quite small, and they did not in the analyses account for confounding factors such as age and comorbidity. The total number studied was 40, of whom just 15 were classified as having the most severe disease. There were some significant differences found, but then this is not surprising given the number of parameters looked at and the number of comparisons made. Perhaps having an initial hypothesis might have reduced the numbers of parameters assessed.

In the statistical analysis of the proteomic data, we accounted for age and sex by marginalization. For routinely performed clinical measurements, marginalization was used as well to account for age and sex, except for the leukocyte count.

Also, we have performed marginalization for the antibody data. For the IgA, the marginalization did not change any significances, while this was not the case for IgG. Therefore, we still present the non-marginalized antibody data for IgG in the manuscript and describe them in the figure legend of Figure 2. Marginalization reports of all analysis are available under <https://github.com/WandrilleD/severe-neuro-COVID-cross-sectional-study-etteretal2022>.

We are aware, that the small samples size is a limitation of this study. However, to our knowledge, it is the biggest prospective COVID-19 cohort study, providing an in depth, multidimensional analysis that includes clinical and paraclinical data. Also, according to Reviewer 2 the number of patients is high compared to other CSF studies of Neuro-COVID.

Particularly, assembling CSF in a prospective manner is quite difficult, since patients with almost no symptoms in most cases decline to undergo lumbar puncture and severely affected patients mostly require anticoagulation, which has to be stopped to perform a lumbar puncture. Furthermore, MRI imaging in intubated and sedated patients and the intensive medicine care setting is logistically challenging.

In our consortial grant proposal for this project, we initially hypothesized that microglia over-activation is important for neurological findings in COVID-19 patients. To elucidate how this is mechanistically happening, we opted for a relatively targeted neuroinflammation proteomics approach in conjunction with structural imaging data, routine clinical data, and, importantly, follow-up exams. We agree that the number of assessed parameters is high. However, our analysis is now able to corroborate the peripheral immune hypothesis with subsequent CNS involvement in a stringent, and well controlled patient cohort.

2. They did well to compare with healthy controls and with inflammatory controls; but it was surprising they did not compare with covid patients who did NOT have CNS disease; without doing this, how do we know to what extent the findings are due to neurological covid disease, as opposed to covid itself?

To make the different findings comparable between different severity states and to clinically contextualize them, we created different Neuro-COVID severity classes. In class I, some patients were asymptomatic (which made it indeed difficult to justify a

lumbar puncture), and the majority suffered only from mild neurological symptoms. In class II, the neurological symptoms were more severe, but still less serious compared to class III, whereas class III patients presented with severe neurological symptoms (e.g. stroke, encephalopathy, seizures). Therefore, we think that the continuum of neurological symptoms with integrative CSF and plasma proteomic patterns can indicate if there exists an association of COVID-19 and neurological symptoms.

3. In terms of numbers of patients, it was surprising so many patients were approached and did not agree to participate. It was unclear on what basis patients were recruited - ie what were the entry criteria to the study. I looked in the Trial protocol which was helpfully included, but I was not really much the wiser (incidentally the protocol said just 20 patients and 20 controls would be recruited, but the numbers went beyond this; I presume there were later amendments which were approved).

Inclusion and exclusion criteria are mentioned under “Clinical characteristics of the study cohorts and study interventions” line 105. In the methods section under “Ethics oversight, patient recruitment, inclusion and exclusion criteria” line 533-536 the recruitment process is described in detail.

From the beginning, we planned on including 20 patients with mild to moderate symptoms and 20 patients with severe symptoms (a total patient population of 40 COVID-19 patients). We wrote amendments to include the University Hospital of Zurich as an additional recruitment site and another amendment to recruit patients from the outpatient clinics testing for COVID-19 there.

4. The paper is quite confusing to follow.

There are subheadings, such as:

“Gray matter volumes in olfactory pathway structures decrease in Neuro-COVID patients and negatively correlate with inflammatory CSF parameters”

But they text then says:

However, this finding was not significant after FDR correction.

Thank you for pointing this out. Indeed, this section is not explained and stated clearly enough to make the reader understand the main message. We therefore rephrased this section to highlight the significant findings, since the negative correlation of specific inflammatory CSF parameters (leukocyte count, protein levels, CSF/plasma albumin ratio) and gray matter volumes was significant ($p < 0.05$).

And:

“High PD-240 L1 and HGF plasma levels are associated with decreased regional gray matter volumes, while GDF-8 and BMP-4 are neuroprotective in Neuro-COVID patients”

But then the text says:

“none of the p-values were significant after BH-procedure.”

If the purported values are not significant after correction, then why the declarative sub-heading?

This is a good point. We changed the subheading to “Lower gray matter volumes in olfactory pathway structures in Neuro-COVID patients are negatively correlated to inflammatory CSF parameters” which now incorporates the significant findings. We mentioned the non-significant findings of these results section at the end and

shortened this part. The corresponding figures were now changed to supplementary figures (Fig. S7-S9). We still feel it is important to report these data since we see a tendency of some biomarkers, e.g. HGF associated with GMV loss in specific brain regions. Larger studies that would target only this biomarker in conjunction with neuroimaging could underscore its importance in COVID-19 neuropathogenesis.

5. The team classified their patients into three groups referencing the paper By Fotuhi et al. However the Fotuhi paper really just comes up with a biological hypothesis about what might be happening in patients, and grade them into three levels, but there is not a great deal of evidence that this hypothesis is correct. The Fotuhi hypothesis assumes a single disease mechanism which explains a continuum of disease severity from I to III, resulting in the different presentations. Although we would all like to believe this, because it makes understanding the disease easier, it may well not be the case! It may be that the disease mechanism causing mild encephalopathy in some patients is completely different to the disease mechanism causing strokes in other patients.

Thank you for this comment. Since COVID-19 is a novel disease entity, we attempted to classify the neurological presentation of our cohort according to severity of the symptoms, taking the Fotuhi paper as a possible source (Fotuhi et al., 2020). It might well be that disease mechanisms do not correlate with the severity grades. As requested by reviewer 2, we also performed additional analysis on COVID-19 severity scores and correlation to our proteomic data, and tried to find common denominators in severe Neuro-COVID and severe COVID, thus enhancing the granularity of our classification.

6. The authors describe the patients as “clinical well-characterised” but unless I missed it, there was very little clinical characterisation. What were the clinical features that led to the patient classification into grades I, II, III? One of the patients with the mildest disease (grade I) died. What happened here? How do they make sense of this?

In the results under “*Clinical characteristics of the study cohorts and study interventions*”, lines 116-123 we described the neurologic symptoms and syndromes that led to the specific classification of our patients. We used the classification criteria of neurological symptoms severity according to Fotuhi et al. and modified these based on our clinical experience. For example, Fotuhi et al. classified tetraplegia presenting during COVID-19 disease as a class II symptom. In contrast, we would have classified a patient presenting with acute tetraplegia as a class III patient. However, mostly the neurological symptom severity classification suggested by Fotuhi et al. was consistent with our opinion on disease severity.

We clarified in the results under “*Long-COVID is more prevalent in severe Neuro-COVID patients and associated with specific CSF and plasma parameters*”, line 355-356 that the deceased class I patient had a preexisting heart failure, which has led to organ failure and death 4 months after COVID-19 infection.

7. The Fotuhi paper does not actually describe how patients should be defined clinically into their three groups, so how did the current authors do this? We get almost no clinical information, such as how many were encephalopathic, how many had strokes, what type, etc etc.

Thank you for pointing this out. We added the information about the main neurological findings per Neuro-COVID class to Table 1. Also, we clarified which class comprised which symptoms and syndromes under “*Clinical characteristics of the study cohorts and study interventions*”, line 116-123. We used the paper of Fotuhi et al. to get an idea

of how to subdivide patients based on their main and most severe neurologic symptom/syndrome. We did not make use of the hypothesis of underlying pathomechanisms that are suggested to be responsible for each class. We made the classification based on the main neurological symptoms at presentation (1) based on suggestions of Fotuhi et al. how to allocate neurologic symptoms to each class and (2) on our own clinical expertise (e.g. patients presenting with tetraplegia were classified as class II patients according to Fotuhi et al. In our opinion, this is a severe neurologic presentation and therefore we would have classified such a patient as a class III patient). In our study, classes comprised symptoms/syndromes as followed: class I: headache, dizziness, loss of smell, loss of taste; class II: myopathy, acute peripheral neuropathy; class III: seizures, neurovascular disorders (stroke, hemorrhage), encephalopathy, coma or death.

In summary, then Etti et al make fabulous use of the samples and imaging they collected, but the numbers are too small to be sure that their findings are meaningful, and the presumption of a severity continuum, as per the hypothesis in the Fotuhi paper limits the usefulness of the interpretation.

- Bellon, M., Schweblin, C., Lambeng, N., Cherpillod, P., Vazquez, J., Lalive, P. H., Schibler, M., & Deffert, C. (2021). Cerebrospinal Fluid Features in Severe Acute Respiratory Syndrome Coronavirus 2 (SARS-CoV-2) Reverse Transcription Polymerase Chain Reaction (RT-PCR) Positive Patients. *Clin Infect Dis*, 73(9), e3102-e3105. <https://doi.org/10.1093/cid/ciaa1165>
- Douaud, G., Lee, S., Alfaro-Almagro, F., Arthofer, C., Wang, C., McCarthy, P., Lange, F., Andersson, J. L. R., Griffanti, L., Duff, E., Jbabdi, S., Taschler, B., Keating, P., Winkler, A. M., Collins, R., Matthews, P. M., Allen, N., Miller, K. L., Nichols, T. E., & Smith, S. M. (2022). SARS-CoV-2 is associated with changes in brain structure in UK Biobank. *medRxiv*. <https://doi.org/10.1101/2021.06.11.21258690>
- Fotuhi, M., Mian, A., Meysami, S., & Raji, C. A. (2020). Neurobiology of COVID-19. *J Alzheimers Dis*, 76(1), 3-19. <https://doi.org/10.3233/jad-200581>
- Heming, M., Li, X., Räuber, S., Mausberg, A. K., Börsch, A. L., Hartlehnert, M., Singhal, A., Lu, I. N., Fleischer, M., Szepanowski, F., Witzke, O., Brenner, T., Dittmer, U., Yosef, N., Kleinschnitz, C., Wiendl, H., Stettner, M., & Meyer Zu Hörste, G. (2021). Neurological Manifestations of COVID-19 Feature T Cell Exhaustion and Dedifferentiated Monocytes in Cerebrospinal Fluid. *Immunity*, 54(1), 164-175.e166. <https://doi.org/10.1016/j.immuni.2020.12.011>
- A minimal common outcome measure set for COVID-19 clinical research. (2020). *Lancet Infect Dis*, 20(8), e192-e197. [https://doi.org/10.1016/s1473-3099\(20\)30483-7](https://doi.org/10.1016/s1473-3099(20)30483-7)
- Neumann, B., Schmidbauer, M. L., Dimitriadis, K., Otto, S., Knier, B., Niesen, W. D., Hosp, J. A., Günther, A., Lindemann, S., Nagy, G., Steinberg, T., Linker, R. A., Hemmer, B., & Bösel, J. (2020). Cerebrospinal fluid findings in COVID-19 patients with neurological symptoms. *J Neurol Sci*, 418, 117090. <https://doi.org/10.1016/j.jns.2020.117090>

Additional changes:

- We changed the Supplementary Figures incorporating the rose plots and heatmaps representing the CSF/plasma ratios of each protein (now Supplementary Figure 3, *rose plots*, and Supplementary Figure 4, *heatmaps*) to better visualize the results. These are now two separate figures. Additionally, the color of the scales and the color of the annotations have been changed and we added the cohorts' annotations at the top of the heatmap using the cohort specific color we used through the whole manuscript.
- We restructured the discussion and have added subtitles so the reader gets a better overview of the different findings and discussion points.

- We updated Supplementary Figure 1 and added the anti-BSA-, anti-dsDNA- and anti-gut bacteria antibody reactivities in the plasma (b) as well as the total antibody levels in the plasma (c)
- We added an excel file containing all the raw data ("file name: source data", supplementary tables). This is stated in the data availability statement on line 781.
- We added a code availability statement including a github link on line 783, providing all statistical reports.

REVIEWERS' COMMENTS

Reviewer #1 (Remarks to the Author):

The authors have revised their manuscript significantly and addressed some of my concerns. However, still they are trying to direct the reader to a causative relationship. I believe this study has several limitations that does not let us make any conclusion regarding the causation effect. For example: "Collectively, these data identified several targets with the potential to prevent COVID-19-related long- and short-term neurological sequelae".

Reviewer #2 (Remarks to the Author):

The authors present a substantially improved manuscript that addresses all my previous concerns and I fully endorse publication.

I am especially impressed by the new mediators identified to be predicting long-COVID. This is an important finding for the field.

I would just recommend the authors to better embed their Long-COVID data with existing literature. They should discuss how their predictors of Long-COVID differ or overlap with previous observation; for example the ones reported by Su et al (PMID 35216672) and Phetsouphanh et al (PMID 35027728).

Reviewer #1 (Remarks to the Author):

The authors have revised their manuscript significantly and addressed some of my concerns. However, still they are trying to direct the reader to a causative relationship. I believe this study has several limitations that does not let us make any conclusion regarding the causation effect. For example: "Collectively, these data identified several targets with the potential to prevent COVID-19-related long- and short-term neurological sequelae".

We thank reviewer 1 for his concerns regarding the causative relationship of our findings in COVID-19 patients. We therefore amended some sentences mainly in the Discussion section to highlight the descriptive manner of the study without implying any causalities. We changed the following statements:

- Discussion section, page 16, line 428: "***We identified Neuro-COVID-specific CSF and plasma alterations, providing insights into possible pathomechanisms underlying COVID-19-related neurological sequelae.***"
- Discussion section, page 23, line 596: "***Collectively, these data identified several possible targets which should be further investigated to potentially prevent COVID-19-related long- and short-term neurological sequelae. Future prospectively designed and larger studies could help to further investigate and assess the causality of our findings and Neuro-COVID severity.***"

With our final remarks in the conclusion part of the Discussion section, we tried to highlight that we present a description of our findings in this manuscript without any implications on the causality of our results and the Neuro-COVID severity.

Reviewer #2 (Remarks to the Author):

The authors present a substantially improved manuscript that addresses all my previous concerns and I fully endorse publication.

I am especially impressed by the new mediators identified to be predicting long-COVID. This is an important finding for the field.

I would just recommend the authors to better embed their Long-COVID data with existing literature. They should discuss how their predictors of Long-COVID differ or overlap with previous observation; for example the ones reported by Su et al (PMID 35216672) and Phetsouphanh et al (PMID 35027728).

We thank reviewer 2 for his feedback and we are glad that we were able to address almost all raised concerns. We are thankful for the constructive suggestion of embedding our long-COVID data with the already existing literature. We followed this advice and integrated the observations described in the 2 suggested papers (Su et al. and Phetsouphanh et al.). The amended long-COVID findings are integrated in the Discussion section on page 22, line 562-564 and on page 23, line 574-579.